# Structural basis for the Rad6 activation by the Bre1 N-terminal domain

**Meng Shi**[1†], **Jiaqi Zhao**[1†], **Simin Zhang**[2†], **Wei Huang**[1], **Mengfei Li**[2], **Xue Bai**[1], **Wenxue Zhang**[3], **Kai Zhang**[1], **Xuefeng Chen**[2]*, **Song Xiang**[1]*

[1]Department of Biochemistry and Molecular Biology, Key Laboratory of Immune Microenvironment and Disease (Ministry of Education), The province and ministry co-sponsored collaborative innovation center for medical epigenetics, Tianjin Medical University, Tianjin, China; [2]Hubei Key Laboratory of Cell Homeostasis, College of Life Sciences, TaiKang Center for Life and Medical Sciences, Frontier Science Centre of Immunology and Metabolism, The Institute of Advanced Studies, Wuhan University, Wuhan, China; [3]Department of Radiation Oncology, Tianjin Medical University General Hospital, Tianjin, China

**Abstract** The mono-ubiquitination of the histone protein H2B (H2Bub1) is a highly conserved histone post-translational modification that plays critical roles in many fundamental processes. In yeast, this modification is catalyzed by the conserved Bre1–Rad6 complex. Bre1 contains a unique N-terminal Rad6-binding domain (RBD), how it interacts with Rad6 and contributes to the H2Bub1 catalysis is unclear. Here, we present crystal structure of the Bre1 RBD–Rad6 complex and structure-guided functional studies. Our structure provides a detailed picture of the interaction between the dimeric Bre1 RBD and a single Rad6 molecule. We further found that the interaction stimulates Rad6's enzymatic activity by allosterically increasing its active site accessibility and likely contribute to the H2Bub1 catalysis through additional mechanisms. In line with these important functions, we found that the interaction is crucial for multiple H2Bub1-regulated processes. Our study provides molecular insights into the H2Bub1 catalysis.

*For correspondence:
xfchen@whu.edu.cn (XC);
xiangsong@tmu.edu.cn (SX)

†These authors contributed equally to this work

Competing interest: The authors declare that no competing interests exist.

## Editor's evaluation

This is a valuable structural study of partial Rad6 from K lactis in complex with Bre1 RBD domain. The structure provides detailed information on the interactions between these two proteins, which are validated by mutagenesis and functional studies. Overall, this is a well-executed study providing solid structural information useful to the field.

## Introduction

The basic unit of the eukaryotic chromatin, nucleosome core particle (NCP), is composed of a protein core consisting of two copies of H2A, H2B, H3, and H4 and double strand DNA wrapped around it (*Luger et al., 1997*). Post-translational modifications (PTMs) on the histone proteins play critical roles in regulating a variety of fundamental processes (*Kouzarides, 2007*; *Strahl and Allis, 2000*). Mono-ubiquitination of a conserved lysine residue in H2B (Lys123 in the budding yeast, Lys120 in mammals, H2Bub1) is a highly conserved histone PTM found in eukaryotes (*West and Bonner, 1980*). It is associated with actively transcribed genes and may contribute to the gene transcription by loosing the chromatin (*Fierz et al., 2011*) and recruiting the FACT histone chaperon complex (*Fleming et al., 2008*; *Pavri et al., 2006*). It also regulates other histone protein PTMs associated with active transcription, including H3K4 and H3K79 methylation catalyzed by DOT1L and the COMPASS complex,

respectively (**Briggs et al., 2002**; **Sun and Allis, 2002**). Recent structural studies indicate that the ubiquitin molecule attached to H2B mediates interactions with DOT1L and the COMPASS complex, contributing to their recruitment and activation (**Anderson et al., 2019**; **Hsu et al., 2019**; **Jang et al., 2019**; **Valencia-Sánchez et al., 2019**; **Worden et al., 2019**; **Worden et al., 2020**; **Xue et al., 2019**; **Yao et al., 2019**). At DNA double strand breaks (DSBs), H2Bub1 facilitates the recruitment of factors for both homologous recombination (HR) and non-homologous end joining pathways, promoting DSB repair (**Moyal et al., 2011**; **Nakamura et al., 2011**; **Shiloh et al., 2011**; **Zheng et al., 2018**). During meiosis, H2Bub1 plays a critical role in the meiotic recombination required for the exchange of genetic materials between homologous chromosomes (**Wang et al., 2017**; **Xu et al., 2016**). H2Bub1 also has important functions in additional processes including DNA replication, nucleosome positioning, RNA processing, chromatin segregation, and others (**Fuchs and Oren, 2014**). In line with its important cellular functions, aberrant H2Bub1 levels in humans are implicated in several types of cancer (**Marsh and Dickson, 2019**; **Marsh et al., 2020**; **Zhou et al., 2021**).

The ubiquitination reaction requires concerted actions of several enzymes. The ubiquitin activating enzyme (E1) conjugates ubiquitin to the ubiquitin-conjugating enzyme (E2), the ubiquitin ligase (E3) subsequently transfers ubiquitin from the E2 enzyme to the substrate. Whereas most organisms contain as few as one or two E1 enzymes and tens of E2 enzymes, they contain a few hundred E3 enzymes, which play critical roles in substrate selection (**Deol et al., 2019**; **Stewart et al., 2016**). In the budding yeast, the H2Bub1 reaction is catalyzed by the E3 enzyme Bre1 together with the E2 enzyme Rad6 (**Hwang et al., 2003**; **Robzyk et al., 2000**; **Wood et al., 2003**). Bre1 is highly conserved among eukaryotes. The human Bre1 orthologs, RNF20 and RNF40, also cooperate with the human Rad6 orthologs (Rad6A and Rad6B) to catalyze the H2Bub1 formation (**Kim et al., 2009**; **Kim et al., 2005**; **Zhu et al., 2005**). Bre1, RNF20, and RNF40 belong to the RING family of E3 enzymes and contain a C-terminal RING domain, which facilitates the ubiquitin transfer to the substrate by promoting the 'closed' conformation of the E2–ubiquitin conjugate (**Zheng and Shabek, 2017**). Bre1's RING domain also interacts with an acidic patch in NCP and positions Rad6 for the H2Bub1 catalysis (**Gallego et al., 2016**). In line these important functions, it was found that removing Bre1's RING domain abolished H2Bub1 in vivo (**Hwang et al., 2003**), and the Bre1 fragment containing the RING domain and a predicted coiled-coil N-terminal to it was able to catalyze H2Bub1 in vitro (**Turco et al., 2015**).

In addition to the RING domain, Bre1 contains a N-terminal Rad6-binding domain (RBD) that also makes important contributions to the H2Bub1 catalysis (**Kim and Roeder, 2009**; **Turco et al., 2015**). However, the mechanism of its interaction with Rad6 and how it contributes to the H2Bub1 catalysis are poorly understood. Here, we present crystal structure of the Bre1 RBD–Rad6 complex and structure-guided functional experiments. Our study revealed detailed mechanism of the Bre1 RBD–Rad6 interaction within a 2:1 Bre1 RBD–Rad6 complex. We found that the interaction stimulates Rad6's enzymatic activity by allosterically increasing its active site accessibility and likely contribute to the H2Bub1 catalysis through additional mechanisms. In line with these important functions, we found that the interaction plays crucial roles in multiple H2Bub1-regulated processes inside the cell.

## Results

### Overall structure of the Bre1 RBD–Rad6 complex

The N-terminal 210 residues in the *Saccharomyces cerevisiae* Bre1 (ScBre1) have been reported to interact with Rad6 and contribute to the H2Bub1 catalysis (**Kim and Roeder, 2009**; **Turco et al., 2015**). This region is highly conserved among fungal Bre1 proteins (**Figure 1—figure supplement 1**). We screened through several fungal species and were able to crystallize the *Kluyveromyces lactis* Rad6 (KlRad6) in complex with the *Kluyveromyces lactis* Bre1 (KlBre1) N-terminal fragment 1-206 or 1-184 (crystal forms 1 and 2, respectively). The Bre1 N-terminal region is predicted to contain several coiled-coils (**Kim and Roeder, 2009**; **Turco et al., 2015**). Using a predicted coiled-coil structure (**Guzenko and Strelkov, 2018**) and the structure of the *Saccharomyces cerevisiae* Rad6 (ScRad6, PDB 1AYZ) (**Worthylake et al., 1998**) as search models, we determined structures of crystal forms 1 and 2 with molecular replacement. The structures were refined to resolutions of 3.2 and 3.05 Å (**Table 1**).

The structures indicate that two KlBre1 N-terminal polypeptides dimerize to form an elongated RBD that binds to one KlRad6 molecule (**Figure 1A**). In both crystal forms, residues 12–173 and 16–182 for KlBre1 polypeptides 1 and 2, respectively, are resolved in the electron density map. Additional KlBre1

**Table 1.** Data collection and structure refinement statistics.

| | Crystal form 1 | Crystal form 2 |
|---|---|---|
| Data collection | | |
| Space group | P6$_5$22 | P6$_1$22 |
| Cell dimensions | | |
| a, b, c (Å) | 94.44, 94.44, 534.64 | 113.22, 113.22, 386.17 |
| α, β, γ (°) | 90, 90, 120 | 90, 90, 120 |
| Resolution (Å) | 50–3.20 (3.26–3.2) | 50–3.05 (3.10–3.05) |
| R$_{merge}$ | 0.200 (1.797) | 0.092 (0.863) |
| I /σI | 17.0 (2.0) | 21.6 (1.1) |
| CC$_{1/2}$ | 0.985 (0.623) | 1.00 (0.876) |
| Completeness (%) | 100.0 (99.0) | 99.7 (96.9) |
| Redundancy | 18.0 (18.5) | 20.6 (16.7) |
| | | |
| Refinement | | |
| Resolution (Å) | 40.89–3.20 (3.35–3.20) | 32.18–3.05 (3.11–3.05) |
| No. reflections | 22,676 (2465) | 28,706 (2495) |
| R$_{work}$ / R$_{free}$ (%) | 21.32/26.33 (32.57/39.34) | 23.72/28.60 (40.29/46.90) |
| No. atoms | | |
| Protein | 7563 | 7486 |
| Ligand/ion | 0 | 17 |
| B-factors | | |
| Protein | 107.63 | 122.41 |
| Ligand/ion | – | 118.27 |
| R.m.s. deviations | | |
| Bond lengths (Å) | 0.003 | 0.003 |
| Bond angles (°) | 0.599 | 0.645 |

Values in parentheses are for the highest resolution shell.

residues included in the expression constructs are presumably disordered. Residues 1–158 in KlRad6 are resolved in the electron density map, part of its acidic C-terminal tail is disordered. Each KlBre1 polypeptide contains two coiled-coil regions (CC1 and CC2), which form extensive parallel coiled-coil interactions with the same regions in the other KlBre1 polypeptide. Each KlBre1 polypeptide also contains a domain M located between CC1 and CC2 that folds CC2 back to interact with CC1. Interactions between CC1 and CC2 and between these coiled-coils and domain M also contribute to the stabilization the RBD structure. The structures of CC1, CC2 and domain M in both KlBre1 polypeptides are quite similar. However, if domain M or CC2 are aligned, the domain M–CC2 linker in the two KlBre1 polypeptides point to opposite directions (*Figure 1B*). There are also structural differences at the N- and C-terminus of the two KlBre1 polypeptides (*Figure 1B*). As a result, the KlBre1 RBD dimer is asymmetric.

Both crystal forms 1 and 2 contain two KlBre1 RBD–Rad6 complexes in the asymmetric unit. The structures of KlBre1 RBD and KlRad6 in these four complexes are similar (*Figure 1—figure supplement 2A, B*), except for regions surrounding KlRad6's active site (detailed below). The structures of KlRad6 in both crystal forms are also similar to the previously reported structure of ScRad6 (*Worthylake et al., 1998*; *Figure 1—figure supplement 2B*). The complexes in both crystal forms adopt similar

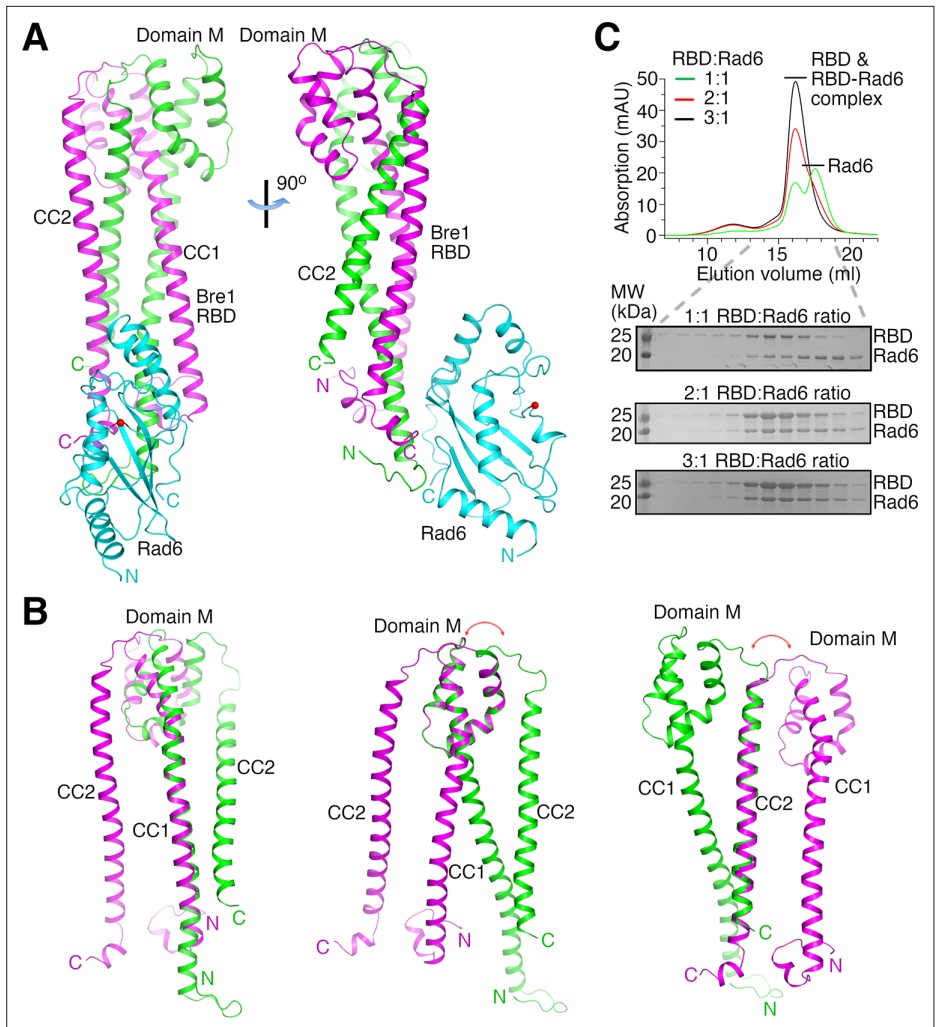

**Figure 1.** Bre1 Rad6-binding domain (RBD) forms a 2:1 complex with Rad6. (**A**) Structure of the KlBre1 RBD–Rad6 complex. The two polypeptides in the KlBre1 RBD are colored in green and magenta, respectively. KlRad6 is colored in cyan. The red spheres indicate the KlRad6 active site. The N- and C-terminus of KlBre1 RBD and KlRad6 are indicated. This coloring scheme is used throughout the manuscript unless indicated otherwise. Structural figures were prepared with PyMOL (https://pymol.org/). (**B**) Structural comparison of the two polypeptides in the KlBre1 RBD. The CC1 (left), domain M (middle), or CC2 (right) regions in these polypeptides are aligned. The red arrows indicate the drastically different orientation of the domain M–CC2 linker in these polypeptides. (**C**) Gel filtration analysis of KlBre1 RBD and KlRad6 mixed at different molar ratios. 15 µM of KlRad was mixed with KlBre1 RBD (1-206) with the indicated molar ratio, injected to a Superdex 200 10/300 column (GE Healthcare) and eluted with a buffer containing 20 mM Tris (pH 7.5) and 200 mM sodium chloride. The lower panels show sodium dodecyl sulfate–polyacrylamide gel electrophoresis (SDS–PAGE) analysis of the gel filtration experiments. Source data for panel C are provided in *Figure 1—source data 1*.

The online version of this article includes the following source data and figure supplement(s) for figure 1:

**Source data 1.** Original gel scan for panel C.

**Figure supplement 1.** Sequence alignment of the N-terminal region in fungal Bre1 proteins.

**Figure supplement 2.** Structure of the KlBre1 Rad6-binding domain (RBD)–Rad6 complexes in the crystal.

**Figure supplement 3.** Cross-linking and gel filtration characterization of KlBre1 Rad6-binding domain (RBD), KlRad6, and their complexes.

**Figure supplement 3—source data 1.** Original gel scans for panels A and B.

**Figure supplement 4.** Structural modeling of Bre1 Rad6-binding domain (RBD) during the H2Bub1 catalysis.

**Figure supplement 5.** E3 enzymes bind to the back side of E2 enzymes with drastically different mechanisms.

conformations, except for complex 1 in crystal form 2. The orientation of KlRad6 in this complex is related to the orientation of KlRad6 in other complexes by a rotation of 14° (*Figure 1—figure supplement 2A*). KlRad6 in this complex forms extensive interactions with neighboring protein molecules in the crystal (*Figure 1—figure supplement 2C*), which may play a role in stabilizing the observed structure. In contrast, KlRad6 molecules in the other three complexes do not mediate extensive crystal packing interactions. Their structure probably better reflect the structure of the KlBre1 RBD–Rad6 complex in solution. We will discuss this structure in the rest of the manuscript.

## Gel filtration characterization of KlBre1 RBD and its complex with Rad6

The dimeric structure of KlBre1 RBD appears to be rather stable, burying 8000 Å² of surface area between the two Bre1 polypeptides. Consistent with the structure, we found that the cross-linking reagent glutaraldehyde can efficiently produce a covalently linked KlBre1 RBD (1-206) dimer (*Figure 1—figure supplement 3A*). To verify that KlBre1 RBD is dimeric in solution and forms a 2:1 complex with KlRad6, we performed gel filtration experiments. We found that the predominant species of KlBre1 RBD (1-206) and the KlBre1 RBD–Rad6 complexes used for crystallization elute at ~16 ml on a Superdex 200 10/300 column (*Figure 1—figure supplement 3B, C*). Calibrating the column with molecules of known sizes indicated an apparent molecular weight of 80 kDa for KlBre1 RBD (1-206) (*Figure 1—figure supplement 3D*), somewhat larger than the expected dimer molecular weight (52 kDa). The elongated shape of the KlBre1 RBD dimer may hinder its interaction with pores

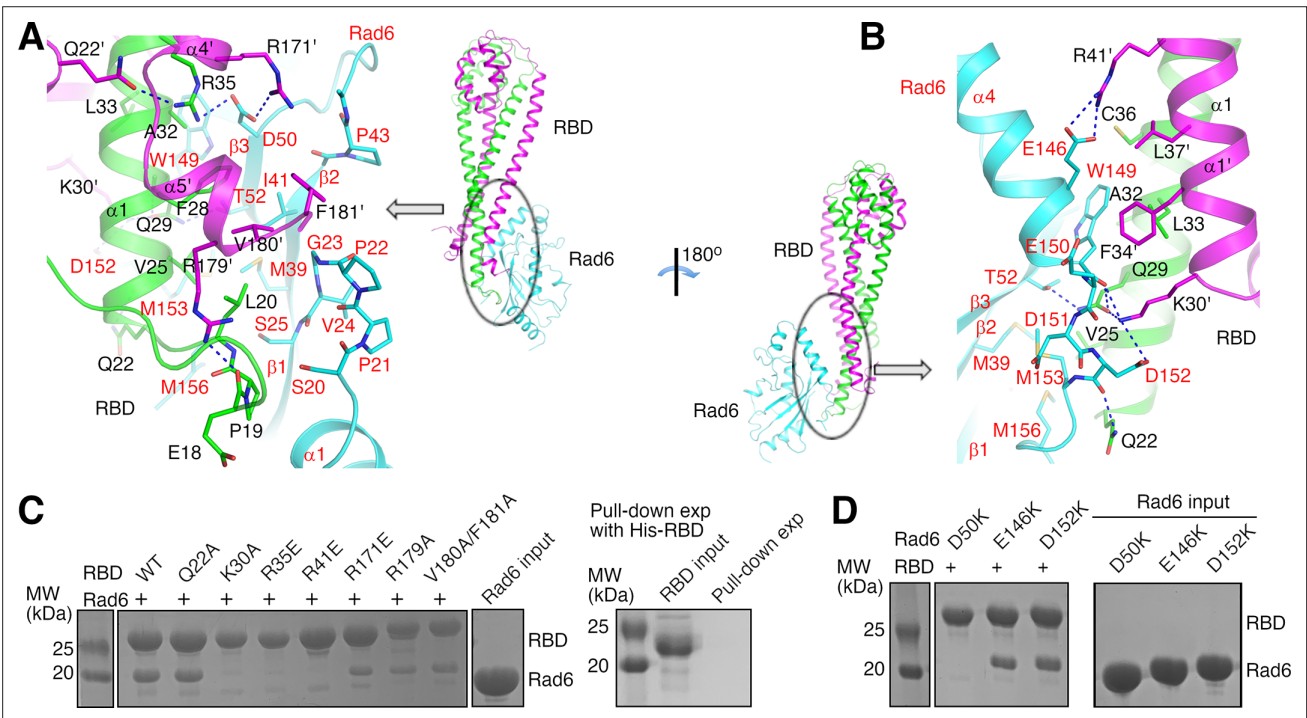

**Figure 2.** Structural basis of the KlBre1 Rad6-binding domain (RBD)–Rad6 interaction. (**A**, **B**) Structure of the KlBre1 RBD–Rad6 interface. Residues important for the interaction are highlighted. The secondary structure elements for KlBre1 RBD are named as in *Figure 1—figure supplement 1*. Labels for KlRad6 are in red. Labels with the prime sign are for the KlBre1 RBD polypeptide 2. (**C**, **D**) KlBre1 RBD–Rad6 pull-down experiments. Sodium dodecyl sulfate–polyacrylamide gel electrophoresis (SDS–PAGE) analysis of strep-tagged KlBre1 RBD (1-206) precipitated with strep-tactin beads and co-precipitated KlRad6 is shown. A control experiment with his-tagged KlBre1 RBD (1-206) is presented in panel C. RBD, KlBre1 RBD. Source data for panels C and D are provided in *Figure 2—source data 1*.

The online version of this article includes the following source data and figure supplement(s) for figure 2:

**Source data 1.** Original gel scans for panels C and D.

**Figure supplement 1.** Residues at the KlBre1 Rad6-binding domain (RBD)–Rad6 interface.

**Figure supplement 2.** Gel filtration characterization of substituted KlBre1 Rad6-binding domain (RBD) (**A**) and KlRad6 (**B**).

**Figure supplement 3.** Isothermal titration calorimetry (ITC) experiments.

**Figure supplement 4.** Surface plasmon resonance (SPR) experiments.

in the gel filtration medium, resulting in its earlier-than-expected elution. The KlBre1 RBD (1-206) sample also contains some minor species that elute earlier. Sodium dodecyl sulfate–polyacrylamide gel electrophoresis (SDS–PAGE) analysis did not reveal any major contaminating proteins (*Figure 1— figure supplement 3B*), suggesting that they represent KlBre1 RBD species that fold differently or contain some minor contaminating proteins. KlRad6 elutes at ~17.5 ml on the same column, consistent with a monomeric form (*Figure 1—figure supplement 3E*). When KlBre1 RBD (1-206) and KlRad6 were mixed with a 1:1 molar ratio and applied to the column, a strong peak of free KlRad6 was observed (*Figure 1C*). In contrast, when they were mixed with 2:1 or 3:1 molar ratio, this peak is absent (*Figure 1C*). Together, these gel filtration experiments are consistent with a dimeric form of KlBre1 RBD in solution and a 2:1 KlBre1 RBD–Rad6 complex.

## Structural basis of the Bre1 RBD–Rad6 interaction

Our structure indicates that KlRad6 interacts with the N-terminal region of KlBre1 RBD polypeptide 1 and N- and C-terminal regions of KlBre1 RBD polypeptide 2, which come together to form a single Rad6-binding site at one end of the RBD. It interacts with the back side of KlRad6 opposite from its active site, composed of strands β1–β3, the C-terminus of α4 and neighboring loops (*Figures 1A and 2A, B*). Part of the C-terminal tail disordered in the structure of the free ScRad6 (*Worthylake et al., 1998*) is structured in KlRad6 and mediates interactions with KlBre1 RBD (*Figure 2B* and *Figure 1— figure supplement 2B*). Residues at the KlBre1 RBD–KlRad6 interface are mostly well defined in the electron density map (*Figure 2—figure supplement 1A, B*). The interface buries 2000 Å$^2$ of surface area and contains hydrophobic, salt bridge and hydrogen bonding interactions (*Figure 2A, B*). At the interface, a hydrophobic patch consisting of KlBre1 residues Pro19, Leu20, Val25, Phe28, Val180' (the prime sign indicates polypeptide 2 in KlBre1 RBD) and Phe181' interacts with KlRad6 residues Ser20-Ser25, Met39, Pro43, Met153, and Met156; a second hydrophobic patch consisting of KlBre1 residues Ala32, Leu33, Cys36, Phe34', and Leu37' interacts with the KlRad6 Trp149 side chain; salt bridges are formed between Arg35 and Arg171' in KlBre1 and Asp50 in KlRad6, Arg41' in KlBre1 and Glu146 in KlRad6, and between Lys30' in KlBre1 and Asp152 in KlRad6; hydrogen bonds are formed between the KlBre1 Gln29 side chain and the KlRad6 Thr52 side chain, the KlBre1 Gln29 and Lys30' side chains and the KlRad6 Trp149 mainchain carbonyl, and the KlBre1 Gln22 side chain and the KlRad6 Asp152 mainchain carbonyl.

Several of the KlBre1 residues at the interface with KlRad6 also contribute to the interactions between the two KlBre1 polypeptides. These residues include Leu20, Leu33, Cys36, Phe34', Leu37', and Val180', which mediate hydrophobic interactions between the two KlBre1 RBD polypeptides; and Gln29, Arg35, and Lys30', which mediate hydrogen bond interactions (*Figure 2A, B*). In addition, several KlBre1 residues contributing to the interaction with KlRad6 in one polypeptide mediate the KlBre1 dimer interactions in the other polypeptide. These residues include Gln22' (*Figure 2A*) and Arg41 (*Figure 2—figure supplement 1C*), which form hydrogen bonds with the Arg35 side chain and the Pro19' mainchain carbonyl, respectively; and Phe34, Leu37, Pro19', Leu33', and Cys36', which mediate hydrophobic interactions (*Figure 2—figure supplement 1C*).

We introduced substitutions at important residues at the KlBre1 RBD–Rad6 interface to probe their function. In KlBre1 RBD (1-206), we introduced charge reversal substitutions R35E, R41E, and R171E to disrupt the observed salt bridge interactions, alanine substitutions at Gln22, Gln29, Lys30, Val180, and Phe181 to disrupt the observed hydrogen bonding or hydrophobic interactions. We also introduced an R179A substitution to abolish the Arg179–Glu18 hydrogen bond, which stabilizes the Pro19–Leu20-containing loop for interaction with KlRad6 (*Figure 2A*). In KlRad6, we introduced the D50K, E146K, and D152K substitutions, which eliminate one of the salt bridges at the interface. The Q29A-substituted KlBre1 RBD (1-206) was poorly expressed and could not be purified. The other substituted KlBre1 RBD (1-206) could be purified to homogeneity. They behave similarly as the wild-type protein in cross-linking (*Figure 1—figure supplement 3A*) and gel filtration experiments (*Figure 2—figure supplement 2A*), suggesting that the substitutions do not significantly alter the dimeric RBD structure, although some of them are located at the interface between the two KlBre1 polypeptides. Gel filtration experiments also indicated that the substitutions in KlRad6 do not significantly alter its overall structure (*Figure 2—figure supplement 2B*).

We next probed the KlBre1 RBD–Rad6 interaction with a pull-down experiment. Streptavidin-binding peptide (strep-) tagged KlBre1 RBD (1-206) was precipitated with strep-tactin beads. When

**Table 2.** Summary of isothermal titration calorimetry (ITC) experiments.

| KlBre1 RBD | KlRad6 | $N$ | $K_D$ (µM) | $\Delta H$ (kcal/mol) | $T\Delta S$ (kcal/mol) | $\Delta G$ (kcal/mol) |
|---|---|---|---|---|---|---|
| With 200 mM salt | | | | | | |
| WT | WT | * | | | | |
| R171E | WT | 0.047 ± 0.366 | 13.9 ± 31.2 | −80 ± 704 | −73.4 | −6.63 |
| With 1 M salt | | | | | | |
| WT | WT | 0.191 ± 0.003 | 0.012 ± 0.009 | −30.0 ± 1.32 | −19.1 | −10.8 |
| Q22A | WT | 0.280 ± 0.02 | 0.207 ± 0.182 | −18.3 ± 2.38 | −9.16 | −9.12 |
| K30A | WT | ND | ND | ND | ND | ND |
| R35E | WT | ND | ND | ND | ND | ND |
| R41E | WT | ND | ND | ND | ND | ND |
| R171E | WT | ND | ND | ND | ND | ND |
| R179A | WT | 0.142 ± 0.007 | 0.819 ± 0.312 | −22.6 ± 2.23 | −14.3 | −8.31 |
| V180A/F181A | WT | 0.193 ± 0.005 | 7.64 ± 0.903 | −31.6 ± 1.78 | −24.6 | −6.98 |
| WT | D50K | ND | ND | ND | ND | ND |
| WT | S111L | 0.289 ± 0.004 | 0.007 ± 0.008 | −33.6 ± 1.52 | −22.5 | −11.1 |
| WT | E146K | 0.134 ± 0.002 | 0.138 ± 0.021 | −32.3 ± 0.63 | −22.9 | −9.37 |
| WT | D152K | 0.134 ± 0.002 | 0.827 ± 0.105 | −30.4 ± 1.02 | −22.1 | −8.30 |

ND, not detectable.
*Unreliable data fitting.

KlRad6 was added to the reaction, a clear co-precipitation was observed (*Figure 2C*). The co-precipitation was abolished by the K30A, R35E, and R41E substitutions in KlBre1 RBD or the D50K substitution in KlRad6, but still present in experiments with the Q22A-, R171E-, R179A-, or V180A/F181A-substituted KlBre1 RBD (1-206) and the wild-type KlRad6, or the wild-type KlBre1 RBD (1-206) and the E146K- or D152K-substituted KlRad6 (*Figure 2C, D*).

To quantify the KlBre1 RBD–Rad6 interaction, we performed isothermal titration calorimetry (ITC) experiments. ITC experiments performed with the same sodium chloride concentration as the pull-down experiments (200 mM) indicated a biphasic binding process (*Figure 2—figure supplement 3A*). However, the heat exchange contributed by the minor phase is too small to enable reliable data fitting with a biphasic binding model but significant enough to interfere with data fitting with a monophasic binding model. We found that increasing the sodium chloride concentration to 1 M suppressed the heat exchange contributed by the minor phase and enabled data fitting with a monophasic binding model, which gave a $K_D$ of 12 nM (*Figure 2—figure supplement 3B* and *Table 2*). The data fitting was not ideal, as the stoichiometry $N$-number is not close to the expected value of 0.5 for a 2:1 binding (*Table 2*). Nevertheless, since the data for this ITC experiment can be fitted, we proceeded to probe the effects of the substitutions with this experiment. Consistent with the pull-down experiments, ITC experiments with 1 M sodium chloride indicated that the K30A, R35E, and R41E substitutions in KlBre1 RBD and the D50K substitution in KlRad6 reduced the affinity to undetectable levels (*Figure 2—figure supplement 3B, C* and *Table 2*). The other substitutions, namely the Q22A, R179A, and V180A/F181A substitutions in KlBre1 RBD and the E146K and D152K substitutions in KlRad6, caused 17- to 640-folds increase in $K_D$. These large increases in $K_D$ indicate that the substitutions inhibited the KlBre1 RBD–Rad6 interaction to various degrees. The interaction between the R171E-substituted KlBre1 RBD (1-206) and KlRad6 is undetectable in ITC experiments with 1 M sodium chloride (*Figure 2—figure supplement 3B* and *Table 2*), but quite substantial in pull-down experiments (*Figure 2C*). We repeated the ITC experiment with 200 mM sodium chloride and found that the interaction was indeed detectable at this condition (*Figure 2—figure supplement 3A* and

**Table 3.** Summary of surface plasmon resonance (SPR) experiments.

| KlBre1 RBD | KlRad6 | $k_a$ (1/Ms) | $k_d$ (1/s) | $K_D$ (μM) |
|---|---|---|---|---|
| WT | WT | $(1.42 \pm 0.002) \times 10^4$ | $(2.05 \pm 0.01) \times 10^{-4}$ | 0.0144 |
| Q22A | WT | $(2.41 \pm 0.009) \times 10^4$ | $(1.52 \pm 0.004) \times 10^{-3}$ | 0.0633 |
| K30A | WT | $(7.88 \pm 0.04) \times 10^2$ | $(3.55 \pm 0.04) \times 10^{-4}$ | 0.451 |
| R35E | WT | ND | ND | ND |
| R41E | WT | ND | ND | ND |
| R171E | WT | ND | ND | ND |
| R179A | WT | $(3.20 \pm 0.02) \times 10^4$ | $(1.66 \pm 0.005) \times 10^{-3}$ | 0.0517 |
| V180A/F181A | WT | * | | 13.7 |
| WT | D50K | ND | ND | ND |
| WT | S111L | $(2.79 \pm 0.005) \times 10^4$ | $(1.47 \pm 0.006) \times 10^{-4}$ | 0.00529 |
| WT | E146K | $(1.82 \pm 0.005) \times 10^4$ | $(1.32 \pm 0.002) \times 10^{-3}$ | 0.0726 |
| WT | D152K | $(2.35 \pm 0.007) \times 10^4$ | $(1.51 \pm 0.003) \times 10^{-3}$ | 0.0641 |

ND, not detectable.
*Steady-state affinity analysis was performed.

*Table 2*). The R171E substitution appears to have a stronger inhibitory effect on the KlBre1 RBD–Rad6 interaction at higher sodium chloride concentration.

To validate the ITC data, we performed surface plasmon resonance (SPR) experiments (*Figure 2—figure supplement 4* and *Table 3*). The SPR experiments indicated that KlBre1 RBD (1-206) binds to immobilized KlRad6 with a $k_a$ of $1.42 \times 10^4$ ($M^{-1} s^{-1}$) and dissociates with a $k_d$ of $2.05 \times 10^{-4}$ ($s^{-1}$). The $K_D$ estimated from these rate constants is 14 nM, similar to the $K_D$ obtained from ITC experiments. Substitutions that strongly inhibited the KlBre1 RBD–Rad6 interaction in the ITC experiments also strongly inhibited the interaction in the SPR experiments. The R35E, R41E, R171E substitutions in KlBre1 RBD and the D50K substitution in KlRad6 reduced the response signal to undetectable levels. The K30A and V180A/F181A substitutions in KlBre1 RBD caused 30- and 950-fold increases in $K_D$, respectively. Additional substitutions, namely Q22A, R179A in KlBre1 RBD and E146K, D152K in KlRad6, have less drastic effects, increasing the $K_D$ 3.6- to 5-fold.

Together the pull-down, ITC and SPR experiments indicate that residues Lys30, Arg35, Arg41 in KlBre1 and Asp50 in KlRad6 play critical roles in the KlBre1 RBD–Rad6 interaction, whereas residues Gln22, Arg171, Arg179, Val180, Phe181 in KlBre1 and Glu146, Asp152 in KlRad6 also make important contributions. Together with the structure, these data provide molecular insights into the KlBre1 RBD–Rad6 interaction.

Previous mutagenesis studies indicated that substitutions on Lys31 in ScBre1 and on Gly23 and Asp50 in ScRad6 abolished their binding (*Turco et al., 2015*). In our structure the equivalent residues, Lys30 in the KlBre1 RBD and Gly23 and Asp50 in KlRad6, mediate important interactions between KlBre1 RBD and KlRad6 (*Figure 2A, B*). Therefore, the reported loss of binding is most likely due to the loss of important interactions at the ScBre1 RBD–Rad6 interface.

## The Bre1 RBD–Rad6 interaction stimulates Rad6's intrinsic activity

Ubiquitin has been reported interact with the back side of several E2 enzymes (*Brzovic et al., 2006*; *Buetow et al., 2015*; *Eddins et al., 2006*; *Hibbert et al., 2011*; *Sakata et al., 2010*). The activity of some of these E2 enzymes, including the human Rad6B, is stimulated by such interaction (*Brzovic et al., 2006*; *Buetow et al., 2015*; *Hibbert et al., 2011*). The expected ubiquitin-binding site on Rad6's back side overlaps with the Bre1 RBD-binding site (*Figure 3A*, left panel), promoting us to probe the effect of Bre1 RBD on Rad6's activity. To this end, we monitored Rad6's intrinsic activity in catalyzing free ubiquitin chain formation (*Hibbert et al., 2011*). To assess the basal activity of Rad6, we introduced a G23R/T52A substitution to KlRad6's back side. The equivalent substitution in the human Rad6B has been reported to abolish ubiquitin binding to its back side

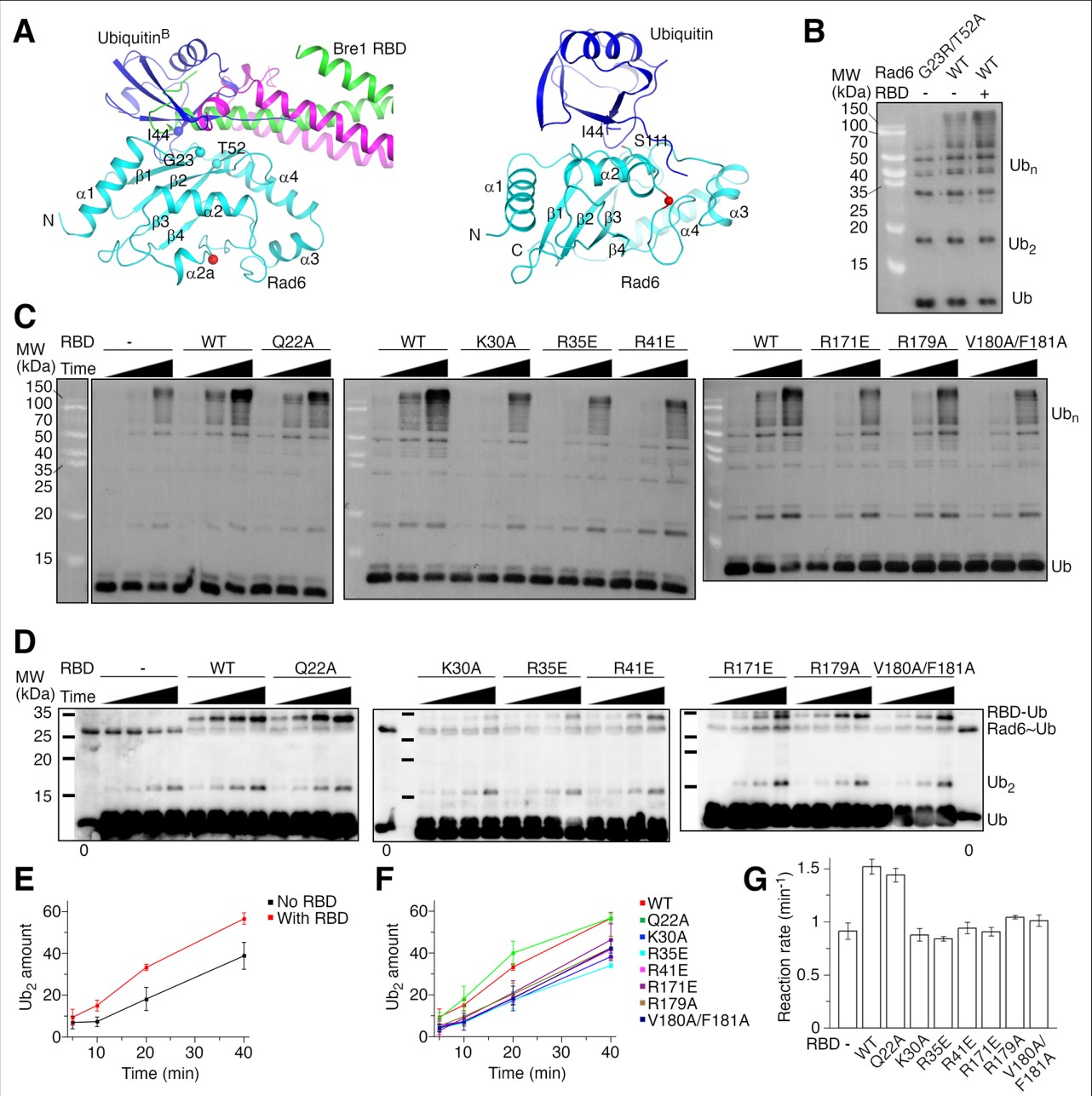

**Figure 3.** KlBre1 Rad6-binding domain (RBD) stimulates KlRad6's activity. (**A**) Structural modeling of the ubiquitin–KlRad6 interaction. The left panel shows a model of KlRad6 with ubiquitin bound at its back side (ubiquitin[B]), based on the structure of UbcH5c with ubiquitin bound at its back side (PDB 2FUH) (***Brzovic et al., 2006***). KlBre1 RBD observed in our structure is shown for reference. The right panel shows a model of the KlRad6–ubiquitin conjugate in the closed conformation, based on the closed conformation of the ubiqtuin–Ubc13 covalent complex (PDB 5ait) (***Branigan et al., 2015***). Secondary structural elements and the N- and C-termini of KlRad6 are indicated. Important residues are highlighted. The red spheres indicate KlRad6's active site. (**B**) Western blot analysis of free ubiquitin chain production by KlRad6. The reactions were carried out for 30 min. KlBre1 RBD (1-206) and ubiquitin chains attached to it were removed before analysis. Ub, ubiquitin; $Ub_2$ and $Ub_n$, ubiquitin chains with 2 or $n$ ubiquitin moieties, respectively. (**C**) Western blot analysis of free ubiquitin chain production in reactions with ubiquitin (I44A) and KlRad6 (S111L). Reactions carried out for 5, 10, and 20 min are presented. KlBre1 RBD (1-206) and ubiquitin chains attached to it were removed before analysis. Two independent repeats of the experiments were performed. (**D**) Western blot analysis of the Rad6–ubiquitin discharging reaction. ubiquitin (I44A/K0) charged to KlRad6 (S111L) was discharged to ubiquitin (I44A) in the absence or presence of KlBre1 RBD (1-206). The reactions were allowed to proceed for 5, 10, 20, and 40 min and analyzed by non-reducing sodium dodecyl sulfate–polyacrylamide gel electrophoresis (SDS–PAGE) followed by western blot for ubiquitin. In lanes marked with '0', western blot analysis of the KlRad6–ubiqtuin conjugate prior to the discharging reaction is presented. Three independent repeats of the experiments were performed. (**E**, **F**) Quantification of the di-ubiquitin production in the KlRad6–ubiquitin discharging reactions. The intensity of the di-ubiquitin band

*Figure 3 continued on next page*

*Figure 3 continued*

divided by the intensity of the Rad6–ubiqtuin band in lane '0' of the same blot times 100 was calculated to represent the di-ubiquitin amount. The error bars represent standard deviations of three independent experiments. Band intensities were read with ImageJ. (**G**) The discharge reaction rate. Data presented in panels E and F were fitted to a linear equation and the slope is used to represent the reaction rate. Errors were derived from data fitting. Source data for panels B–D are provided in *Figure 3—source data 1*, for panels E–G are provided in *Figure 3—source data 2*.

The online version of this article includes the following source data and figure supplement(s) for figure 3:

**Source data 1.** Original gel scans for panels B-D.

**Source data 2.** Data points for panels E-G.

**Figure supplement 1.** Rad6 activity assays.

**Figure supplement 1—source data 1.** Original gel scans and blots for panels A-G.

**Figure supplement 1—source data 2.** Data points for panel H.

and the ubiquitin-mediated stimulation (*Hibbert et al., 2011*). Gel filtration experiments suggest that the substitution did not alter the overall structure of KlRad6 (*Figure 2—figure supplement 2B*). Consistent with ubiquitin binding to the back side of KlRad6 and stimulating its activity, we found that the substitution decreased the free ubiquitin chain production by KlRad6 (*Figure 3B* and *Figure 3—figure supplement 1A*). Supplementing KlBre1 RBD (1-206) to the reaction with the wild-type KlRad6 accelerated the formation of ubiquitin chains (*Figure 3—figure supplement 1B*), but the ubiquitin chains can be attached to KlBre1 RBD (1-206) (*Figure 3—figure supplement 1C*). After removing the strep-tagged KlBre1 RBD (1-206) and attached ubiquitin chains with strep-tactin beads (*Figure 3—figure supplement 1B*), we found that the reaction with the wild-type KlRad6 and KlBre1 RBD (1-206) also produced more free ubiquitin chains than the reaction with KlRad6 (G23R/T52A) (*Figure 3B*). The affinity between ubiquitin and Rad6 is rather weak, with $K_D$ in the mM range (*Kumar et al., 2015*). In contrast, our binding experiments indicated a strong interaction between KlBre1 RBD and KlRad6. Therefore, when KlBre1 RBD is supplemented to the reaction, KlBre1 RBD but not ubiquitin is expected to occupy KlRad6's back side. Thus, our experiments indicate that KlBre1 RBD stimulates KlRad6's intrinsic activity.

KlRad6 residues Gly23 and Thr52 are located at the center of its interface with KlBre1 RBD (*Figure 2A, B*) and we found that the G23R/T52A substitution severely inhibits the KlBre1 RBD–Rad6 interaction (*Figure 3—figure supplement 1D*). Hence, KlRad6 (G23R/T52A) is not expected to be activated by KlBre1 RBD. To compare the basal and Bre1 RBD-stimulated activities of the same Rad6 variant, we introduced the I44A substitution to ubiquitin. This substitution inhibits the interaction between ubiquitin and the back side of E2 enzymes (*Figure 3A*, left panel; *Brzovic et al., 2006*; *Buetow et al., 2015*; *Eddins et al., 2006*; *Hibbert et al., 2011*; *Sakata et al., 2010*). It also desta-bilizes the closed conformation of the E2–ubiquitin conjugate required for ubiquitin transfer, but this defect can be rescued by substitutions in the E2 enzyme near the active site (*Li et al., 2015*; *Saha et al., 2011*). One of such substitutions, equivalent to S111L in KlRad6, introduces hydrophobic inter-actions with the substituted I44A side chain to stabilize the closed conformation (*Figure 3A*, right panel). The S111L substitution did not inhibit the KlBre1 RBD–Rad6 interaction (*Figure 3—figure supplement 1D*, *Tables 2 and 3*) or alter the overall structure of KlRad6 (*Figure 2—figure supplement 2B*). As expected, it rescued the severely inhibited free ubiquitin chain production associated with ubiquitin (I44A) (*Figure 3—figure supplement 1A*). Supplementing KlBre1 RBD (1-206) into the reaction with KlRad6 (S111L) and ubiquitin (I44A) stimulated the free ubiquitin chain production, consistent with a stimulatory effect of KlBre1 RBD on KlRad6's intrinsic activity (*Figure 3C*).

To further probe the role of the KlBre1 RBD–Rad6 interaction in the stimulation of KlRad6's activity, we tested the effects of substitutions in KlBre1 RBD. In free ubiquitin chain formation reactions with KlRad6 (S111L) and ubiquitin (I44A), we found that the Q22A substitution did not noticeably change the stimulation by KlBre1 RBD (1-206), whereas substitutions K30A, R35E, R41E, R171E, R179A, and V180A/F181A decreased stimulation (*Figure 3C*). These observations correlate well with our binding experiments, which indicate that the Q22A substitution moderately inhibited the KlBre1 RBD–KlRad6 interaction whereas the other substitutions caused stronger inhibitions (*Figure 2C*, *Tables 2 and 3*). Together, these data indicate a critical role of the Bre1 RBD–Rad6 interaction in stimulating Rad6's intrinsic activity.

## The Bre1 RBD–Rad6 interaction stimulates Rad6's activity in ubiquitin discharging

The ubiquitination reaction consists of two steps, ubiquitin charging to the E2 enzyme and ubiquitin discharging from the E2–ubiquitin conjugate to the substrate. Ubiquitin and several E3 enzymes have been reported to bind to the back side of E2 enzymes to regulate the discharging reaction (*Buetow et al., 2015*; *Das et al., 2009*; *Li et al., 2015*; *Metzger et al., 2013*). To test if the Bre1 RBD–Rad6 interaction affects ubiquitin discharging from the Rad6–ubiquitin conjugate, we performed single turnover ubiquitin discharging experiments. After charging ubiquitin to KlRad6 with the E1 enzyme, we stopped the charging reaction and allowed the KlRad6–ubiquitin conjugate to discharge. We found that the amount of the conjugate decreases over time and adding KlBre1 RBD (1-184) accelerated the rate of its disappearance (*Figure 3—figure supplement 1E*), in line with a previous report (*Turco et al., 2015*). Since KlBre1 RBD itself can be ubiquitinated, adding it could accelerate the ubiquitin discharge by providing more acceptor substrates. To eliminate this effect and test if KlBre1 RBD regulates KlRad6's enzymatic activity in ubiquitin discharging, we followed the reaction of ubiquitin discharging to ubiquitin (*Figure 3D–G*). The substrate amount of this reaction is unaffected by supplementing KlBre1 RBD. We monitored the amount of the reaction product, the free ubiquitin chain. To better quantify the reaction, we utilized a K0-subsituted donor ubiquitin in which all lysine residues are substituted to arginine. After its attachment to the acceptor ubiquitin, donor ubiquitin attachment to it is prevented, making di-ubiquitin the only product. To eliminate the potential regulatory effect of ubiquitin by binding to KlRad6's back side, we introduced the I44A and S111L substitutions to ubiquitin and KlRad6, respectively. Western blot analysis indicated that the di-ubiquitin production in this reaction is stimulated by supplementing the wild-type KlBre1 RBD (1-206) (*Figure 3D, E, G*). Similar to the free ubiquitin chain production experiments, we found that the Q22A substitution in KlBre1 RBD had little effect on the stimulation, whereas substitutions K30A, R35E, R41E, R171E, R179A, and V180A/F181A decreased the stimulation (*Figure 3D, F, G*). Together, these data indicate that the KlBre1 RBD–Rad6 interaction stimulates KlRad6's enzymatic activity in ubiquitin discharging.

KlBre1 RBD (1-206) interferes with the western blot detection of the KlRad6–ubiquitin conjugate (*Figure 3D* and *Figure 3—figure supplement 1F*). The conjugate can be effectively detected with SDS–PAGE with 18% gels (*Figure 3—figure supplement 1F*) and we quantified its amount with this technique. The data show that KlBre1 RBD (1-206) also stimulated the disappearance of the KlRad6–ubiquitin conjugate in the above discharging experiments (*Figure 3—figure supplement 1G, H*). The stimulation is reduced by substitutions in KlBre1 RBD inhibiting the KlBre1 RBD–Rad6 interaction, with the smallest reduction observed for the Q22A substitution and the largest reductions observed for the K30A, R35E and R41E substitutions. Consistent with ubiquitin discharging to KlBre1 RBD, we found an appearance of the KlBre1 RBD–ubiquitin product in the discharging experiments (*Figure 3D*). The rate of its appearance is reduced by the K30A, R35E, and R41E substitutions in KlBre1 RBD (*Figure 3D*), suggesting that a strong KlBre1 RBD–Rad6 interaction is required for efficient ubiquitin discharge to KlBre1 RBD. Together, data from di-ubiquitin, KlBre1 RBD–ubiquitin and the KlRad6–ubiquitin conjugate are consistent with the notion that both ubiquitin discharge to ubiquitin and to KlBre1 RBD contributes to the disappearance of the KlRad6–ubiquitin conjugate.

## Bre1 RBD increases Rad6's active site accessibility

Structural analysis provided insights into the mechanism of the observed Rad6 stimulation. When the three KlRad6 molecules not mediating extensive crystal packing interactions in our structures are compared, large structural differences are observed for residues 90–98 and 115–121 surrounding KlRad6's active site (*Figure 4A*). The 90–98 region can be resolved in the electron density map in crystal form 1 but is mostly disordered in KlRad6 molecule 2 in crystal form 2, the equivalent Cα atoms in the 115–121 region are located as far as 5.2 Å apart. These regions in the three KlRad6 molecules also display significantly elevated temperature factors (*Figure 4B*). In contrast, in KlRad6 molecule 1 in crystal form 2, the temperature factors for these regions are not elevated (*Figure 4—figure supplement 1A*), consistent with stabilization of their structure by the extensive crystal packing interactions (*Figure 1—figure supplement 2C*). The equivalent regions in Rad6 orthologs have been reported to be mobile. NMR studies indicated that these regions in the human Rad6B are flexible on a nanosecond to picosecond time scale (*Miura et al., 1999*; *Miura et al., 2002*). Structural differences were also observed for these regions in the crystal structure of the closely related ScRad6 (sharing 94.5%

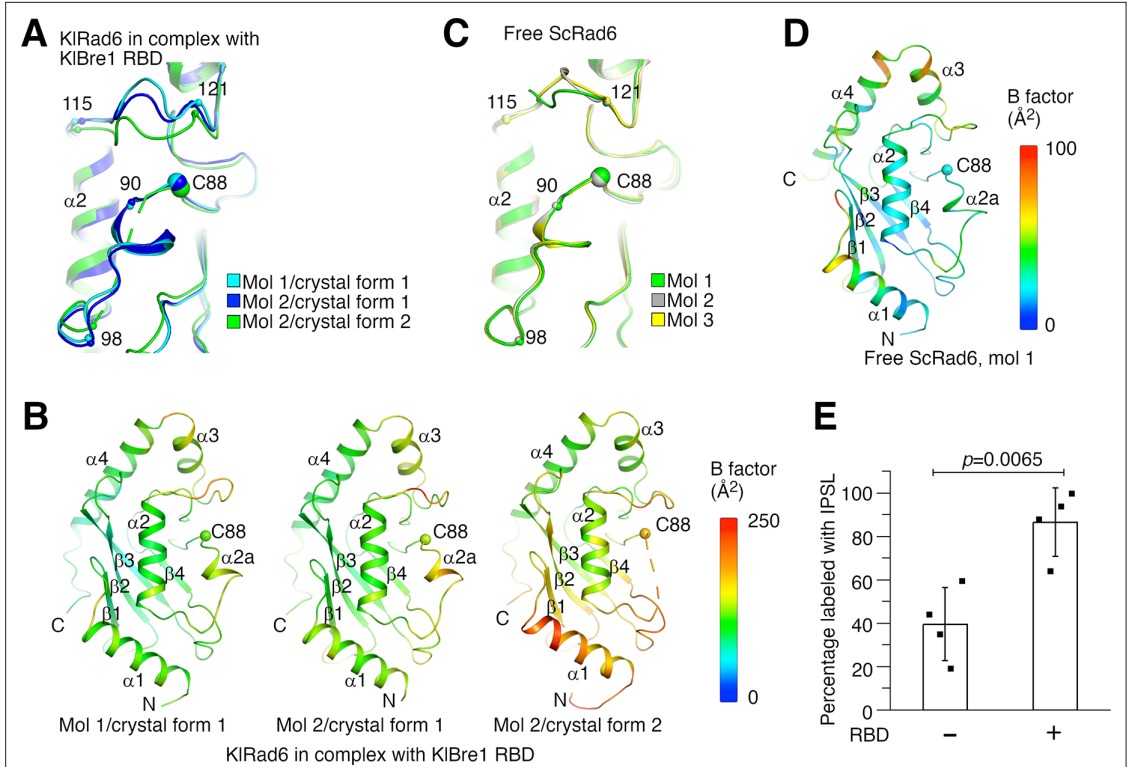

**Figure 4.** Bre1 Rad6-binding domain (RBD) increases Rad6's active site accessibility. (**A**) Structural comparison of KlRad6 molecules in our crystals. Regions surrounding the active site (C88) in KlRad6 molecules not mediating extensive crystal packing interactions are shown. The Cα positions of residues 90, 98, 115, and 121 are indicated. (**B**) Temperature factor distribution of KlRad6 molecules in our crystals. The structures of KlRad6 molecules not mediating extensive crystal packing interactions are colored according to the temperature factor. The average temperature factor for regions 90–98 and 115–121 in KlRad6 molecules 1 and 2 in crystal form 1 and molecule 2 in crystal form 2 are 143.7, 156.4, and 173.5 Å², respectively, the average temperature factor for the rest of these molecules are 104.6, 117.1, and 149.2 Å². (**C**) Structural comparison of ScRad6 molecules in the crystal structure of free ScRad6 (PDB 1AYZ) (**Worthylake et al., 1998**). (**D**) Temperature factor distribution of ScRad6 in the crystal structure of free ScRad6. Molecule 1 in the crystal is presented. The average temperature factor for its regions 90–98 and 115–121 is 45.7 Å², for the rest of the molecule is 42.6 Å². (**E**) KlBre1 RBD increase KlRad6's active site accessibility. 3-(2-Iodoacetamido)-2,2,5,5-tetramethyl-1-pyrolidinyloxy (IPSL) labeling of KlRad6's active site cysteine is presented. The error bars represent standard deviations of four independent experiments (represented by the black dots). The p value is derived from the two-sided Student's t-test. Source data for panel E are provided in **Figure 4—source data 1**.

The online version of this article includes the following source data and figure supplement(s) for figure 4:

**Source data 1.** Data points for panel E.

**Figure supplement 1.** Temperature factor distribution of Rad6 molecules in crystal structures.

sequence identity with KlRad6) (**Worthylake et al., 1998**), but are less pronounced. In the three ScRad6 molecules in the crystal, the 90–98 region adopt almost identical structures, the equivalent Cα atoms in the 115–121 region are located less than 4.1 Å apart (**Figure 4C**). The temperature factors of these regions are also elevated in this structure, but less significantly (**Figure 4D** and **Figure 4—figure supplement 1B**). Together, these data indicate an increased mobility of regions 90–98 and 115–121 in KlRad6 in the KlBre1 RBD–Rad6 complex, suggesting that KlBre1 RBD allosterically increase the mobility of these regions. Such an increase in mobility is expected to make KlRad6's active site more accessible, which could facilitate the nucleophilic attack of the KlRad6–ubiquitin thioester bond and promotes the ubiquitin discharge.

To test if KlBre1 RBD increases KlRad6's active site accessibility, we labeled its active site cysteine with the cysteine-reacting reagent 3-(2-Iodoacetamido)-2,2,5,5-tetramethyl-1-pyrolidinyloxy (IPSL). We found that after a 15-min incubation with 300 µM IPSL, 39% of KlRad6's active site cysteine was labeled with IPSL, supplementing KlBre1 RBD (1-206) increased the labeling level to 86% (**Figure 4E**). Together with the structural data, these data indicate that KlBre1 RBD allosterically increases KlRad6's active site accessibility to stimulate its activity.

## The Bre1 RBD–Rad6 interaction is crucial for Bre1's function inside the cell

To assess the physiological function of the Bre1 RBD–Rad6 interaction inside the cell, we generated a *BRE1* knock out yeast strain and complemented it with the wild-type or substituted ScBre1. The effects of several ScBre1 substitutions were assessed, including Q23A, Q30A, K31D, R36E, R42E, and the Q30A/K31D and R36E/R42E double substitutions. In line with previous reports (*Hwang et al., 2003*; *Wood et al., 2003*), we found that knocking out *BRE1* abolished H2Bub1 in vivo (*Figure 5A*), caused sensitivity toward replication stress or DNA damaging reagents including camptothecin (CPT), methylmethane sulfonate (MMS), phleomycin, and hydroxy urea (HU) (*Figure 5B*). These phenotypes can be rescued by complementing the cells with the wild-type or Q23A-substituted ScBre1, but not ScBre1 with the other substitutions. The Bre1-mediated H2Bub1 plays a critical role in promoting DNA DSB repair by HR (*Zheng et al., 2018*). To test the importance of the Bre1 RBD–Rad6 interaction in HR, we employed an ectopic recombination system in which a single DSB is generated by the HO endonuclease at the *MAT*a sequence inserted in chromosome V, which can be repaired by HR using the homologous *MAT*a-inc sequence located on chromosome III (*Figure 5C*; *Ira et al., 2003*). Approximately 55% of the *BRE1* knock out cells completed the repair and survived. Complementing the cells with the wild-type or the Q23A-substituted ScBre1 increased the survival rate to ~80%, while complementing the cells with ScBre1 carrying the other substitutions failed to or only modestly increased the survival rate (55–72%) (*Figure 5D*). These substitutions did not significantly alter the Bre1 or Rad6 protein levels (*Figure 5E*). These data are in line with our structure and biochemical studies. The Q23A substitution did not cause noticeable defects in the cellular H2Bub1 level, sensitivity toward replication stress or DNA damaging reagents or the HR repair. The equivalent substitution in KlBre1 RBD, Q22A, only moderately reduced its affinity toward KlRad6. In contrast, strong defects were observed for the Q30A, K31D, R36E, R42E substitutions or their combinations. KlBre1 residues equivalent to the substituted ones mediate important hydrogen bond or salt bridge interactions with Rad6 (Gln29, Lys30, Arg35, and Arg41, *Figure 2A, B*), and our binding experiments indicated that substitutions on Lys30, Arg35, and Arg41 strongly inhibited the KlBre1 RBD–KlRad6 interaction (*Figure 2C*, *Tables 2 and 3*). The strong defects associated with these substitutions correlate with the expected strong inhibition of the ScBre1 RBD–Rad6 interaction. Together, these data indicate that the Bre1 RBD–Rad6 interaction is crucial for the Bre1-mediated H2B mono-ubiquitination, DNA damage response, and HR repair inside the cell.

## Discussion

Our structure provides mechanistic insights into the interaction between the two Bre1 RBD polypeptides and between Bre1 and Rad6. It has been reported that Bre1 forms a dimer and its N-terminal region plays a critical role in the dimer formation (*Kim and Roeder, 2009*). In line with this report, our structure indicates that multiple regions in the two Bre1 N-terminal polypeptides contribute to a large dimer interface. Our structure also indicates that multiple regions in both Bre1 polypeptides contribute to the interaction with Rad6. Such a structure suggests that the Bre1 dimerization is required for the Bre1 RBD–Rad6 interaction.

An important finding we made is that the Bre1 RBD–Rad6 interaction stimulates Rad6's enzymatic activity. Our data suggest that by binding to Rad6's back side, Bre1 RBD allosterically increases Rad6's activity site accessibility, which facilitats the nucleophilic attack of the Rad6–ubiqtuin thioester bond to promote ubiquitin discharge. Ubiquitin also binds to the back side of several E2 enzymes to stimulate their activity (*Brzovic et al., 2006*; *Buetow et al., 2015*; *Hibbert et al., 2011*). However, there are striking differences between the Bre1 RBD- and ubiquitin-mediated stimulation. Unlike Bre1 RBD, ubiquitin does not form stable complexes with E2 enzymes and it is not clear if it allosterically increases their active site accessibility. In addition to this mechanism, our data suggest that the Bre1 RBD–Rad6 interaction may promote the H2Bub1 catalysis through additional mechanisms. First, it may maintain a close Bre1–Rad6 association during the H2Bub1 catalysis. Structural studies indicated that the E1 enzyme and the RING domain in the E3 enzymes compete for binding to the E2 enzyme (*Cappadocia and Lima, 2018*; *Gundogdu and Walden, 2019*; *Stewart et al., 2016*; *Streich and Lima, 2014*), resulting a 'ping-pong' motion of the E2 enzyme between the E1 enzyme and the RING domain during the catalysis. The Bre1 RBD–Rad6 interaction is not expected to interfere

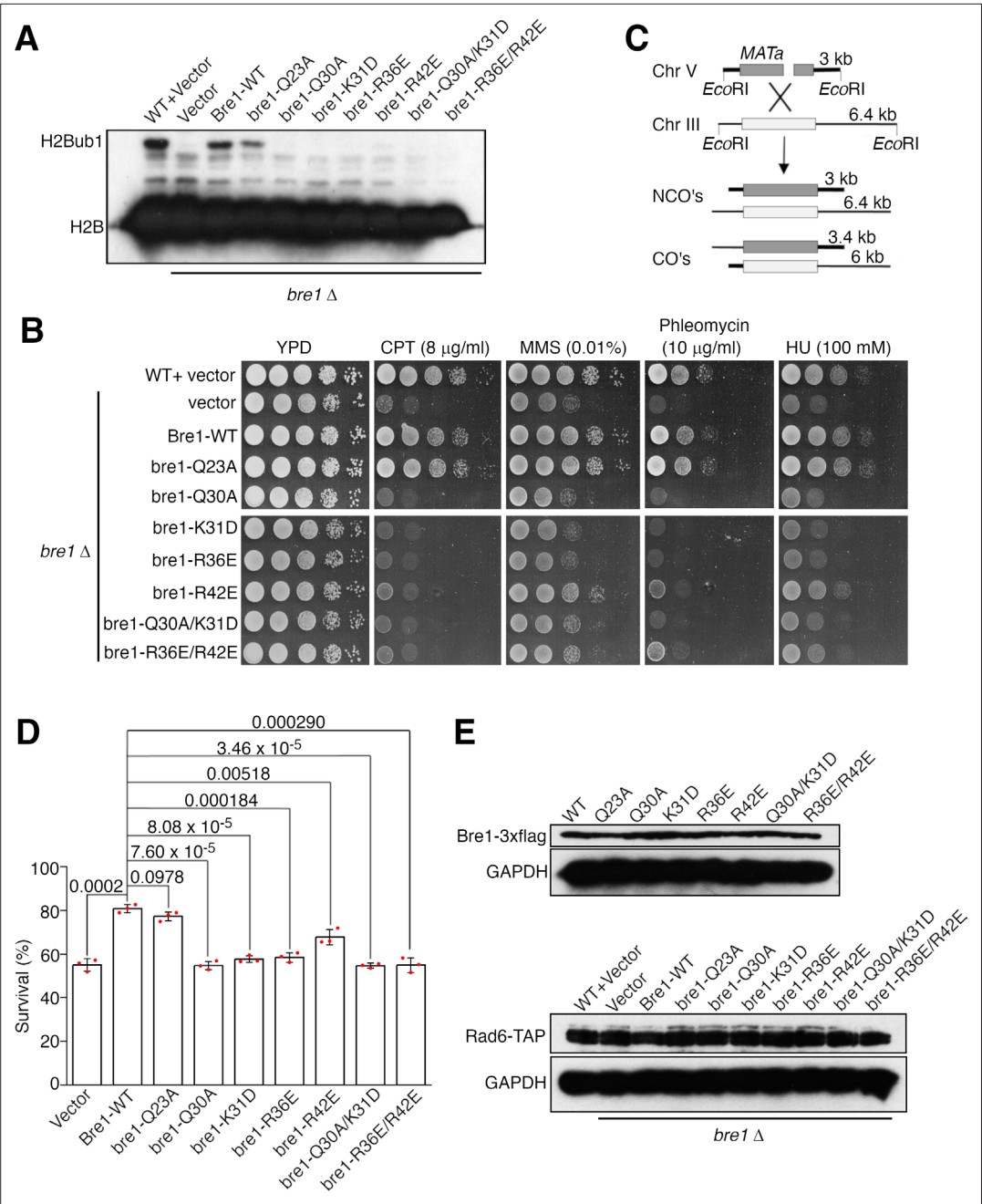

**Figure 5.** The Bre1 Rad6-binding domain (RBD)–Rad6 interaction is crucial for Bre1's function inside the cell. (**A**) The Bre1 RBD–Rad6 interaction is essential for the H2Bub1 formation inside the cell. Western blot analysis for H2Bub1 in *BRE1* knock out cells complemented with pRS316-derived plasmids carrying the wild-type or substituted ScBre1 is shown. Data for the wild-type or *BRE1* knock out cells complemented with the empty vector are included for comparison. (**B**) The Bre1 RBD–Rad6 interaction is essential for survival in the presence of replication stress/DNA damaging reagents. Sensitivity of *BRE1* knock out cells complemented with pRS16-derived plasmids for the wild-type or substituted ScBre1 is shown. The cells were challenged with camptothecin (CPT), methylmethane sulfonate (MMS), phleomycin, or hydroxy urea (HU). Data for the wild-type or *BRE1* knock out cells complemented with the empty vector are included for comparison. (**C**) Scheme showing the ectopic recombination system. CO, crossover; NCO, non-crossover. (**D**) The Bre1 RBD–Rad6 interaction is essential for the HR repair. Survival rate of *BRE1* knock out cells complemented with the wild-type or substituted ScBre1 due to successful HR repair is shown. Data for cells complemented with the empty vector are included for comparison. Error bars represent standard deviations of three independent experiments (represented by the red dots). p values derived from the two-sided Student's *t*-test are presented. (**E**) Substitutions in ScBre1 do not alter the Bre1 or Rad6

*Figure 5 continued on next page*

*Figure 5 continued*

protein levels. Western blot analysis for Bre1 and Rad6 is presented. Western blot analysis for glyceraldehyde-3-phosphate dehydrogenase (GAPDH) is included as reference. Source data for panels A and E are provided in *Figure 5—source data 1*, for panel D is provided in *Figure 5—source data 2*.

The online version of this article includes the following source data for figure 5:

**Source data 1.** Original blots for panels A and E.

**Source data 2.** Data points for panel D.

with the Rad6–E1 or Rad6–RING interactions (*Figure 1—figure supplement 4A, B*). Therefore, the strong affinity between Bre1 RBD and Rad6 we and others (*Turco et al., 2015*) observed suggests a constant Rad6–Bre1 association during the ubiquitination catalysis. Such a Rad6–Bre1 association could increase the local concentration of Rad6 to promote the catalysis. Second, the Bre1 RBD–Rad6 interaction may position Bre1 RBD for interaction with ubiquitin in the Rad6–ubiquitin conjugate and may stabilize its closed conformation for ubiquitin discharge (*Figure 1—figure supplement 4C*). The conserved phenylalanine equivalent to Phe181 in KlBre1 is expected to play a central role in this interaction. The loss of this interaction may contribute to the observed loss of KlRad6 stimulation by the V180A/F181A-substituted KlBre1 RBD (*Figure 3C, D, F, G*). These important functions of the Bre1 RBD–Rad6 interaction are in line with our observation that it is crucial for multiple H2Bub1-regulated processes inside the cell.

In a recent study, it was found that supplementing ScBre1 RBD to a single turnover discharging experiment stimulates ubiquitin discharge from the Rad6–ubiquitin conjugate (*Turco et al., 2015*). In addition to ubiquitin discharging to ubiquitin (reaction 1), our data suggest that supplementing ScBre1 RBD probably introduced a second reaction, ubiquitin discharging to ScBre1 RBD (reaction 2). Reaction 1 is relevant to Rad6's enzymatic activity and the H2Bub1 catalysis in vivo. Whether reaction 2 takes place in vivo is not clear. Both reactions contribute to the Rad6–ubiquitin discharge, making it difficult to deduce from the accelerated Rad6–ubiquitin discharge whether ScBre1 RBD stimulates reaction 1. In addition, our structure suggests that Bre1 RBD competes with ubiquitin for binding to Rad6's back side and eliminates the ubiquitin-mediated stimulation on Rad6. Therefore, measuring Rad6's basal activity without ubiquitin bound at its back side is necessary to reveal its regulation by Bre1 RBD. In our study, we monitored reaction 1 and introduced substitutions to access Rad6's basal activity. We found that the reaction 1 rate is stimulated by supplementing KlBre1 RBD and the stimulation is inhibited by substitutions that inhibits the KlBre1 RBD–Rad6 interaction. Such data provide strong evidence that the Bre1 RBD–Rad6 interaction stimulates Rad6's enzymatic activity. It should be noted that in the presence of Bre1 RBD, reaction 2 inhibits reaction 1 by competing with it for the Rad6–ubiquitin conjugate. Therefore, the increase in reaction 1 rate by Bre1 RBD represents an underestimation of its stimulatory effect.

Several other E3 enzymes have been reported to contain specific E2 enzyme back side binding regions (E2BBRs), which regulate the activity of related E2 enzymes with distinct mechanisms. E2BBRs in Gp78 (*Das et al., 2013*; *Das et al., 2009*) and Cue1 *Metzger et al., 2013* have been reported to stimulate the activity of the related E2 enzymes by increasing their active site accessibility and interaction with the RING domain, whereas E2BBRs in AO7 (*Li et al., 2015*) and Rad18 *Hibbert et al., 2011* have been reported to inhibit the activity of the related E2 enzymes by blocking ubiquitin binding to their back side. All the previously reported E2BBRs possess a single E2-interacting region that contains a major α helix, which binds to the back side of E2 enzymes in different orientations (*Figure 1—figure supplement 5*). In sharp contrast to these E2BBRs, multiple regions in both polypeptides in the Bre1 RBD dimer contribute to the interaction with Rad6. Most interestingly, although both the Bre1 RBD and the Rad18 E2BBR bind to the Rad6 back side, they appear to have opposite effects on Rad6's activity. Together, these previously studies and ours indicate that E3 enzymes can interact with the back side of E2 enzymes with drastically different mechanisms to regulate their activity.

In the Bre1 holoenzyme, RBD coordinates with other Bre1 domains and additional factors to catalyze the H2Bub1 reaction. Our study provided glimpse into this important reaction, yet extensive future studies are required to fully understand its mechanism. One of the most important questions to answer is how Bre1 RBD coordinates with its RING domain during the H2Bub1 catalysis. Unlike Bre1 RBD, the Bre1 RING domain appears to be a symmetrical dimer with two Rad6-binding sites

(*Kumar and Wolberger, 2015*). Studies suggest that only one of them interacts with the Rad6–ubiquitin conjugate for ubiquitin transfer to NCP (*Gallego et al., 2016*). It is also important to understand the coordination between Bre1 RBD and additional domains/factors during the H2Bub1 catalysis. For instance, the non-canonical back side of Rad6 appears to be accessible in the Bre1 RBD-bound Rad6 (*Figure 1—figure supplement 4B*), where ubiquitin may bind to regulate its activity (*Kumar et al., 2015*). A clear understanding of the H2Bub1 catalytic mechanism will help to resolve the long-standing puzzle how Bre1 directs Rad6's activity toward substrate mono-ubiquitination. Rad6 participates in both ubiquitin chain formation and mono-ubiquitination reactions. It functions with Ubr1 to catalyze the ubiquitin chain modification of N-end rule protein substrates (*Dohmen et al., 1991*), with Bre1 (*Hwang et al., 2003*; *Robzyk et al., 2000*; *Wood et al., 2003*) and Rad18 (*Hoege et al., 2002*) to mono-ubiquitinate H2B and the proliferating cell nuclear antigen (PCNA), respectively. It also possesses an intrinsic activity to catalyze free ubiquitin chain formation (*Hibbert et al., 2011*). The Rad18 E2BBR suppresses its intrinsic free ubiquitin chain forming activity and directs its activity toward PCNA mono-ubiquitination (*Hibbert et al., 2011*). In contrast, our data indicate that Bre1 RBD does not inhibit the free ubiquitin chain formation by Rad6 but stimulates its activity. Thus, Bre1 utilizes a completely different mechanism to direct Rad6's activity toward H2B mono-ubiquitination. Understanding this mechanism could provide insights into the catalytic mechanism of E3 enzymes in general.

The human orthologs of Bre1, RNF20, and RNF40, form a heterodimer (*Kim et al., 2009*). It has been reported that the RNF20 N-terminal 381 residues could form a complex with RNF40 that binds Rad6 (*Kim et al., 2009*), suggesting that these residues participate in the formation of a functional RBD. However, sequence analysis suggested that the region spanning residues 328–528 in RNF20 possesses homology to the RBD region in Bre1 (*Zhu et al., 2005*). Therefore, future studies are required to define the RBD in the RNF20/RNF40 heterodimer and elucidate its structure and function.

# Materials and methods

## Key resources table

| Reagent type (species) or resource | Designation | Source or reference | Identifiers | Additional information |
|---|---|---|---|---|
| Strain, strain background (*Saccharomyces cerevisiae*) | JKM139 | *Lee et al., 1998* | | |
| Strain, strain background (*Saccharomyces cerevisiae*) | tGI354 | *Ira et al., 2003* | | |
| Strain, strain background (*Saccharomyces cerevisiae*) | yZSH241 | This study | bre1::TRP1 | Parental strain: tGI354 |
| Strain, strain background (*Saccharomyces cerevisiae*) | yMF001 | This study | bre1::TRP1 + pRS316-bre1-Q23A | Parental strain: tGI354 |
| Strain, strain background (*Saccharomyces cerevisiae*) | yMF002 | This study | bre1::TRP1 + pRS316-bre1-Q30AK31D | Parental strain: tGI354 |
| Strain, strain background (*Saccharomyces cerevisiae*) | yMF003 | This study | bre1::TRP1 + pRS316-bre1-R36ER42E | Parental strain: tGI354 |
| Strain, strain background (*Saccharomyces cerevisiae*) | yMF004 | This study | bre1::TRP1 + pRS316 | Parental strain: tGI354 |
| Strain, strain background (*Saccharomyces cerevisiae*) | yMF005 | This study | bre1::TRP1 + pRS316-BRE1 | Parental strain: tGI354 |
| Strain, strain background (*Saccharomyces cerevisiae*) | yMF006 | This study | bre1::TRP1 + pRS316-bre1-Q23A | Parental strain: JKM139 |
| Strain, strain background (*Saccharomyces cerevisiae*) | yMF007 | This study | bre1::TRP1 + pRS316-bre1-Q30AK31D | Parental strain: JKM139 |
| Strain, strain background (*Saccharomyces cerevisiae*) | yMF008 | This study | bre1::TRP1 + pRS316-bre1-R36ER42E | Parental strain: JKM139 |
| Strain, strain background (*Saccharomyces cerevisiae*) | yMF009 | This study | bre1::TRP1 + pRS316 | Parental strain: JKM139 |

*Continued on next page*

*Continued*

| Reagent type (species) or resource | Designation | Source or reference | Identifiers | Additional information |
|---|---|---|---|---|
| Strain, strain background (*Saccharomyces cerevisiae*) | yMF010 | This study | *bre1::TRP1 + pRS316-BRE1* | Parental strain: JKM139 |
| Strain, strain background (*Saccharomyces cerevisiae*) | yLD002 | This study | *bre1::TRP1* | Parental strain: JKM139 |
| Strain, strain background (*Saccharomyces cerevisiae*) | yMF011 | This study | *KanMX-FLAG-H2B bre1::TRP1 + pRS316-bre1-Q23A* | Parental strain: JKM139 |
| Strain, strain background (*Saccharomyces cerevisiae*) | yMF012 | This study | *KanMX-FLAG-H2B bre1::TRP1 + pRS316-bre1-Q30AK31D* | Parental strain: JKM139 |
| Strain, strain background (*Saccharomyces cerevisiae*) | yMF013 | This study | *KanMX-FLAG-H2B bre1::TRP1 + pRS316-bre1-R36ER42E* | Parental strain: JKM139 |
| Strain, strain background (*Saccharomyces cerevisiae*) | yMF014 | This study | *KanMX-FLAG-H2B bre1::TRP1 + pRS316* | Parental strain: JKM139 |
| Strain, strain background (*Saccharomyces cerevisiae*) | yMF015 | This study | *KanMX-FLAG-H2B bre1::TRP1 + pRS316-BRE1* | Parental strain: JKM139 |
| Strain, strain background (*Saccharomyces cerevisiae*) | yGX060 | This study | *KanMX-FLAG-H2B bre1::TRP1* | Parental strain: JKM139 |
| Strain, strain background (*Saccharomyces cerevisiae*) | yZS059 | This study | *bre1::TRP1 + pRS316-bre1-Q30A* | Parental strain: tGI354 |
| Strain, strain background (*Saccharomyces cerevisiae*) | yZS060 | This study | *bre1::TRP1 + pRS316-bre1-K31D* | Parental strain: tGI354 |
| Strain, strain background (*Saccharomyces cerevisiae*) | yZS061 | This study | *bre1::TRP1 + pRS316-bre1-R36E* | Parental strain: tGI354 |
| Strain, strain background (*Saccharomyces cerevisiae*) | yZS062 | This study | *bre1::TRP1 + pRS316-bre1-R42E* | Parental strain: tGI354 |
| Strain, strain background (*Saccharomyces cerevisiae*) | yZS063 | This study | *bre1::TRP1 + pRS316-bre1-Q30A* | Parental strain: JKM139 |
| Strain, strain background (*Saccharomyces cerevisiae*) | yZS064 | This study | *bre1::TRP1 + pRS316-bre1-K31D* | Parental strain: JKM139 |
| Strain, strain background (*Saccharomyces cerevisiae*) | yZS065 | This study | *bre1::TRP1 + pRS316-bre1-R36E* | Parental strain: JKM139 |
| Strain, strain background (*Saccharomyces cerevisiae*) | yZS066 | This study | *bre1::TRP1 + pRS316-bre1-R42E* | Parental strain: JKM139 |
| Strain, strain background (*Saccharomyces cerevisiae*) | yZS067 | This study | *bre1::TRP1 + pRS316-BRE1-3FLAG* | Parental strain: JKM139 |
| Strain, strain background (*Saccharomyces cerevisiae*) | yZS068 | This study | *bre1::TRP1 + pRS316-bre1-Q23A-3FLAG* | Parental strain: JKM139 |
| Strain, strain background (*Saccharomyces cerevisiae*) | yZS069 | This study | *bre1::TRP1 + pRS316-bre1-Q30A-3FLAG* | Parental strain: JKM139 |
| Strain, strain background (*Saccharomyces cerevisiae*) | yZS070 | This study | *bre1::TRP1 + pRS316-bre1-K31D-3FLAG* | Parental strain: JKM139 |
| Strain, strain background (*Saccharomyces cerevisiae*) | yZS071 | This study | *bre1::TRP1 + pRS316-bre1-R36E-3FLAG* | Parental strain: JKM139 |
| Strain, strain background (*Saccharomyces cerevisiae*) | yZS072 | This study | *bre1::TRP1 + pRS316-bre1-R42E-3FLAG* | Parental strain: JKM139 |
| Strain, strain background (*Saccharomyces cerevisiae*) | yZS073 | This study | *bre1::TRP1 + pRS316-bre1-Q30AK31D-3FLAG* | Parental strain: JKM139 |
| Strain, strain background (*Saccharomyces cerevisiae*) | yZS074 | This study | *bre1::TRP1 + pRS316-bre1-R36ER42E-3FLAG* | Parental strain: JKM139 |
| Strain, strain background (*Saccharomyces cerevisiae*) | yZS075 | Dharmacon (Yeast TAP Tagged ORFs) | *RAD6-TAP-HIS* | Parental strain: By4741 |

*Continued on next page*

*Continued*

| Reagent type (species) or resource | Designation | Source or reference | Identifiers | Additional information |
|---|---|---|---|---|
| Strain, strain background (*Saccharomyces cerevisiae*) | yZS076 | This study | *RAD6-TAP-HIS bre1::KanMX* | Parental strain: By4741 |
| Strain, strain background (*Saccharomyces cerevisiae*) | yZS077 | This study | *RAD6-TAP-HIS+pRS316* | Parental strain: By4741 |
| Strain, strain background (*Saccharomyces cerevisiae*) | yZS078 | This study | *RAD6-TAP-HIS bre1::KanMX + pRS316* | Parental strain: By4741 |
| Strain, strain background (*Saccharomyces cerevisiae*) | yZS079 | This study | *RAD6-TAP-HIS bre1::KanMX + pRS316-BRE1* | Parental strain: By4741 |
| Strain, strain background (*Saccharomyces cerevisiae*) | yZS080 | This study | *RAD6-TAP-HIS bre1::KanMX + pRS316-bre1-Q23A* | Parental strain: By4741 |
| Strain, strain background (*Saccharomyces cerevisiae*) | yZS081 | This study | *RAD6-TAP-HIS bre1::KanMX + pRS316-bre1-Q30A* | Parental strain: By4741 |
| Strain, strain background (*Saccharomyces cerevisiae*) | yZS082 | This study | *RAD6-TAP-HIS bre1::KanMX + pRS316-bre1-K31D* | Parental strain: By4741 |
| Strain, strain background (*Saccharomyces cerevisiae*) | yZS083 | This study | *RAD6-TAP-HIS bre1::KanMX + pRS316-bre1-R36E* | Parental strain: By4741 |
| Strain, strain background (*Saccharomyces cerevisiae*) | yZS084 | This study | *RAD6-TAP-HIS bre1::KanMX + pRS316-bre1-R42E* | Parental strain: By4741 |
| Strain, strain background (*Saccharomyces cerevisiae*) | yZS085 | This study | *RAD6-TAP-HIS bre1::KanMX + pRS316-bre1-Q30AK31D* | Parental strain: By4741 |
| Strain, strain background (*Saccharomyces cerevisiae*) | yZS086 | This study | *RAD6-TAP-HIS bre1::KanMX + pRS316-bre1-R36ER42E* | Parental strain: By4741 |
| Antibody | anti-ubiquitin (Mouse monoclonal) | Santa Cruz Biotechnology | Cat# sc-8017, RRID: AB_628423 | 1:2000 |
| Antibody | anti-strep (Mouse monoclonal) | ABclonal | Cat# AE066, RRID: AB_2863792 | 1:5000 |
| Antibody | anti-FLAG (Mouse monoclonal) | Sigma-Aldrich | Cat# F3165, RRID: AB_259529 | 1:3000 |
| Antibody | anti-FLAG (Mouse monoclonal) | Sigma-Aldrich | Cat# F1804, RRID: AB_262044 | 1:3000 |
| Antibody | anti-TAP (Mouse monoclonal) | Zen-BioScience | Cat# 250067 | 1:6000 |
| Antibody | anti-GAPDH (Mouse monoclonal) | ABclonal | Cat# AC033, RRID: AB_2769570 | 1:50,000 |
| Peptide, recombinant protein | HRP-conjugated mouse IgGκ light chain binding protein | Santa Cruz Biotechnology | Cat# sc-516102, RRID: AB_2687626 | 1:6000 |

## Protein expression and purification

The coding regions for KlBre1 fragment 1-206 or 1-184 and the full-length KlRad6 were amplified from the *Kluyveromyces lactis* genome and inserted into vectors pET26B and pET28A (Novagen), respectively. The recombinant KlBre1 fragments and KlRad6 contain no tags and an N-terminal 6x histidine (his-) tag, respectively. For protein expression, *Escherichia coli* BL21 Rosetta (DE3) cells harboring these plasmids were induced with 0.2 mM isopropyl β-D-1-thiogalactopyranoside (Bio Basic) for 16 hr at 16°C. For complex purification, cells expressing one of the KlBre1 fragments were mixed with cells expressing KlRad6 and lysed with an AH-2010 homogenizer (ATS Engineering). The KlBre1 RBD–Rad6 complex was purified by nickel–nitrilotriacetic acid agarose (Ni-NTA, Smart Life sciences) and ion exchange (Hitrap Q HP, GE Healthcare) columns, followed by a 2-hr treatment with thrombin (5 units for 1 mg of complex) at room temperature to remove the his-tag on KlRad6, and further purified by gel filtration (Superdex 200 10/300, GE Healthcare). Purified complexes were concentrated to 10 mg/ml in a buffer containing 20 mM Tris (pH 7.5), 200 mM sodium chloride, and 2 mM dithiothreitol (DTT), flashed cooled in liquid nitrogen and stored at −80°C.

Unless otherwise indicated, KlBre1 RBD (1-206) containing N-terminal his- and strep-tags and KlRad6 containing N-terminal his- and hemagglutinin (HA-) tags were used for the biochemical assays. The coding regions for the KlBre1 fragment and the full length KlRad6 were inserted into

vector pET28A, and the strep- and HA-tags were introduced by Polymerase Chain Reaction (PCR)-based mutagenesis. The recombinant double tagged KlBre1 RBD (1-206) and KlRad6 proteins were expressed in BL21 Rosetta (DE3) cells and purified with Ni-NTA, Heparin (Hitrap Heparin HP, GE Healthcare) and gel filtration (Superdex 200 10/300, GE Healthcare) columns. To purify his-tagged KlBre1 RBD (1-206) or (1-184), the corresponding KlBre1 gene fragment was inserted into vector pET28A. The recombinant protein was expressed in BL21 Rosetta (DE3) cells and purified by Ni-NTA, Heparin (Hitrap Heparin HP) and gel filtration (Superdex 200 10/300) columns. The *S. cerevisiae* E1 protein Uba1 and ubiquitin were purified as described (*Lee and Schindelin, 2008*; *Shen et al., 2021*). Briefly, the gene fragment corresponding to Uba1 residues 10–1024 was inserted into vector pET28A. Uba1 was expressed in BL21 Rosetta (DE3) cells and purified with Ni-NTA, hydrophobic interaction (Hitrap Butyl HP, GE Healthcare) and gel filtration (Superdex 200 Increase 10/300) columns. The ubiquitin gene was inserted into vector pET28A. Ubiquitin was expressed in in BL21 Rosetta (DE3) cells and purified with Ni-NTA, ion-exchange (Hitrap Q HP) and gel filtration (Superdex 200 10/300) columns. To purify untagged KlRad6 and ubiquitin, the his-tagged proteins were incubated with thrombin (5 units for 1 mg of protein) at room temperature for 2 hr, followed by purification with gel filtration (Superdex 200 increase 10/300) column.

Amino acid substitutions were introduced with a PCR-based protocol and verified by DNA sequencing. The substituted proteins were expressed and purified following the same protocols for the wild-type proteins.

The selenomethionine (SeMet)-substituted KlRad6 was produced by inhibiting the host methionine production and supplementing SeMet (*Doublié et al., 1996*). The KlBre1 RBD (1-184)–Rad6 complex containing SeMet-substituted KlRad6 was purified following the same protocol for the native complex, except that the DTT concentration is increased to 10 mM in the storage buffer.

## Crystallization and structural determination

Hexagon-shaped crystals of the complex containing the KlBre1 RBD (1-206) and KlRad6 (crystal form 1) were obtained with vapor diffusion sitting drop experiments at 18°C. The reservoir solution contains 0.2 M sodium acetate (pH 5.5) and 15% PEG3350. Before data collection, the crystals were equilibrated in the reservoir solution supplemented with 25% glycerol, flash cooled and stored in liquid nitrogen. Diffraction data were collected at the Shanghai Synchrotron Radiation Facility (SSRF) Beamline BL19U1 at 0.97853 Å on a Pilatus 6M detector. Diffraction data were indexed, integrated, and scaled with the HKL3000 suite (*Otwinowski and Minor, 1997*). Initial molecular replacement calculations with PHASER (*McCoy et al., 2007*) with the structure of ScRad6 (PDB 1AYZ) (*Worthylake et al., 1998*) as the search model did not yield interpretable electron density maps. An analysis with PAIRCOIL2 (*McDonnell et al., 2006*) indicated that the KlBre1 N-terminal region contains at least one coiled-coil of 40 residues in length. A predicted coiled-coil structure was generated with CCFOLD (*Guzenko and Strelkov, 2018*). Using this structure and the structure of ScRad6 as search models, molecular replacement calculations with PHASER and subsequent density improvement with PHENIX (*Adams et al., 2010*) produced interpretable density maps. Model building was carried out with O (*Jones et al., 1991*) and COOT (*Emsley and Cowtan, 2004*). Refinement was carried out with PHENIX.

Elongated diamond-shaped crystals of the complex containing the KlBre1 RBD (1-184) and SeMet-substituted KlRad6 (crystal form 2) were obtained with vapor diffusion sitting drop experiments at 18°C. The reservoir solution contains 0.1 M sodium acetate (pH 4.6) and 1 M ammonium sulfate. Diffraction data were collected on SSRF beamline BL19U1 at 0.97846 Å and processed with the HKL3000 suite. The structure was determined with molecular replacement with PHASER, using the structure of crystal form 1 as the search model. It was refined with PHENIX.

## Cross-linking experiments

For cross-linking, 30 μM of KlBre1 RBD (1-206) was incubated with 2% glutaraldehyde in a buffer containing 20 mM Tris (pH 7.5) and 200 mM sodium chloride for 30 min at 37°C. The reaction was analyzed with SDS–PAGE.

## Pull-down experiments

To probe the KlBre1 RBD–Rad6 interaction, 15 μM his-strep-tagged KlBre1 RBD (1-206) was incubated with 4 μM KlRad6 in a binding buffer containing 20 mM Tris (pH 7.5) and 200 mM sodium chloride on

ice for 30 min. The reaction mixture was subsequently incubated with 30 µl strep-tactin beads (Smart Lifesciences) equilibrated in the binding buffer for 2 hr at 4°C. After washing the beads twice with the binding buffer, bound proteins were eluted with the binding buffer supplemented with 2 mM desbiotin and analyzed with SDS–PAGE.

## Isothermal titration calorimetry

ITC experiments were performed on a MicroCal PEAQ-ITC instrument (Malvern) at 25°C. Prior to the ITC experiments, both KlBre1 RBD (1-206) and KlRad6 were exchanged in a buffer containing 20 mM Tris (pH 7.5) and sodium chloride at indicated concentrations. To characterize binding, a solution containing 100 µM (for experiments with the wild-type or Q22A-, K30A-, R35E-, R41E-, R171E-, and R179A-substituted KlBre1 RBD with 1 M sodium chloride or the wild-type KlBre1 RBD with 200 mM sodium chloride) or 150 µM (for the experiment with the S111L-subsbituted KlRad6) or 200 µM (for the experiment with the R171E-substitued KlBre1 RBD with 200 mM sodium chloride) or 400 µM (for the experiment with the V180A/F181A-substituted KlBre1 RBD) KlRad6 was injected into a 300 µl cell that stores 50 µM KlBre1 RBD (1-206), 2 µl at a time. Data were analyzed with ORIGIN 7.0 (Originlab).

## Surface plasmon resonance

SPR experiments were performed on a Biacore 8K instrument (Cytiva) at 25°C with a flow rate of 30 µl/min. To immobilized KlRad6 on a CM5 chip, KlRad6 was first diluted with PBS buffer (10 mM phosphate buffer (pH 7.4), 2.7 mM potassium chloride, 137 mM sodium chloride) to 100 µg/ml and subsequently with 10 mM sodium acetate (pH 4.5) to 1–3 µg/ml and flown over the chip for 60 s. To characterize KlBre1 RBD binding, KlBre1 RBD (1-206) was diluted with the HEPES buffer (20 mM HEPES [pH 7.5], 200 mM sodium chloride) to the indicated concentrations and flown over the chip for 120 s. KlBre1 RBD was subsequently dissociated from the chip by a 600- (for the experiment with the S111L-substituted KlRad6) or 300- (for other experiments) s wash with the HEPES buffer. After each experiment, the chip was regenerated by flowing 10 mM glycine–HCl (pH 2.75) over the chip for 30 s. Data were analyzed with the Biacore Insight Evaluation software (Cytia).

## IPSL labeling

For IPSL labeling, 10 µM KlRad6 was incubated with 300 µM IPSL for 15 min in a buffer containing 20 mM Tris (pH 7.5) and 200 mM sodium chloride, in the presence or absence of 40 µM KlBre1 RBD. The reaction was stopped by adding 600 µM L-cysteine. The level of IPSL labeling was accessed by mass spectrometry (MS). After tryptic digestion of the reaction mixture, the resulting peptides were separated by nano-liquid chromatography on an easy-nLC 1200 system (Thermo Fisher Scientific) and directly sprayed into a Q-Exactive Plus mass spectrometer (Thermo Fisher Scientific). The MS analysis was carried out in data-dependent mode with an automatic switch between a full MS and a tandem MS (MS/MS) scan in the Orbitrap. For full MS survey scan, the automatic gain control target was set to 1e6, and the scan range was from 350 to 1750 with a resolution of 70,000. The 10 most intense peaks with charge state ≥2 were selected for fragmentation by higher energy collision dissociation with normalized collision energy of 27%. The MS2 spectra were acquired with a resolution of 17,500, and the exclusion window was set at ±2.2 Da. All MS/MS spectra were searched using the PD search engine (v 1.4.0, Thermo Fisher Scientific) with an overall false discovery rate for peptides less than 1%. Peptide sequences were searched using trypsin specificity allowing a maximum of two missed cleavages. IPSL-Alkylation (+198.137 Da) on cysteine, acetylationon peptide N-terminal and oxidation of methionine were specified as variable modifications. Mass tolerances for precursor ions were set at ±10 ppm for precursor ions and ±0.02 Da for MS/MS. The ratio of labeled peptides to unlabeled peptides in the mass spectrometric analysis was calculated to represent the level of IPSL labeling.

## Free ubiquitin chain formation

The reaction mixture contains 60 mM Tris (pH 8.5), 50 mM sodium chloride, 50 mM potassium chloride, 10 mM magnesium chloride, 0.1 mM DTT, 3 mM ATP, 90 nM Uba1, 60 µM ubiquitin, and 10 µM (for experiments presented in *Figure 3B* and *Figure 3—figure supplement 1B, C*) or 5 µM KlRad6 (for experiments presented in *Figure 3C* and *Figure 3—figure supplement 1A*). When indicated, 20 µM KlBre1 RBD (1-206) was supplemented. After incubating at 30°C for the indicated period, the reactions were stopped by boiling in the SDS–PAGE loading buffer. To remove KlBre1 RBD, the

boiled reaction mixture was subject to strep-tactin beads precipitation twice and the supernatant was collected for analysis. For comparison, reactions without KlBre1 RBD were supplemented with the same amount of KlBre1 RBD after the reaction and subjected to the same treatment with strep-tactin beads. The reaction mixtures were analyzed with SDS–PAGE and western blot with an anti-ubiquitin antibody (sc-8017, Santa Cruz Biotechnology, RRID: AB_628423, 1:2000 diluted). To probe ubiquitin chains attached to the KlBre1 RBD during the reaction, reaction mixtures prior to the treatment with strep-tactin beads were analyzed with western blot with an anti-strep antibody (AE066, ABclonal, RRID: AB_2863792, 1:5000 diluted).

## Rad6–ubiquitin discharging

For the discharging reactions, untagged KlRad6 and ubiquitin were used. For discharging reactions with the wild-type KlRad, KlRad6 was first charged with ubiquitin with a reaction mixture containing 60 mM Tris (pH 8.0), 50 mM sodium chloride, 50 mM potassium chloride, 10 mM magnesium chloride, 0.1 mM DTT, 3 mM ATP, 30 nM Uba1, 50 μM ubiquitin, and 2 μM KlRad6. After a 10-min incubation at 30°C, the charging reaction was stopped by adding 50 mM Ethylenediaminetetraacetic acid (EDTA) (pH 8.0) and ubiquitin discharge from the KlRad6–ubiqtuin conjugate in the presence or absence of 20 μM KlBre1 RBD (1-184) was allowed to proceed at 30°C. After the indicated period, aliquots of the reaction were removed and analyzed by non-reducing SDS–PAGE. For discharging reactions with KlRad6 (S111L), KlRad6 (S111L) was first charged with ubiquitin (I44A/K0) with a reaction mixture containing 60 mM Tris (pH 8.0), 50 mM sodium chloride, 50 mM potassium chloride, 10 mM magnesium chloride, 0.1 mM DTT, 3 mM ATP, 300 nM Uba1, 20 μM ubiquitin (I44A/K0), and 10 μM KlRad6 (S111L). The reaction mixture was incubated at 30°C for 30 min, with additional 300 nM Uba1 and 3 mM ATP supplemented at minutes 10 and 20. The charging reaction was stopped by adding 50 mM EDTA (pH 8.0) and the discharging reaction was initiated by supplementing 60 μM ubiquitin (I44A) to the reaction mixture. When indicated, 30 μM wild-type or substituted KlBre1 RBD (1-206) was supplemented. The reaction mixture was incubated at 30°C for the indicated period and analyzed by non-reducing SDS–PAGE or non-reducing SDS–PAGE followed by western blot with an anti-ubiquitin antibody (sc-8017, Santa Cruz Biotechnology, RRID: AB_628423, 1:2000 diluted). For quantification, band intensities were read by ImageJ (https://imagej.nih.gov/ij/).

## Yeast strains and plasmids

Yeast strains used are listed in the Key Resources Table. All strains used in this study are derivates of JKM139 (*ho MATa hml::ADE1 hmr::ADE1 ade1-100 leu2-3,112 trp1::hisG' lys5 ura3-52 ade3::GAL::HO*) or tGI354 (*MATa-inc arg5,6::MATa-HPH ade3::GAL::HO hmr::ADE1 hml::ADE1 ura3-52,* for ectopic recombination tests) or BY4741 (*MATa his3Δ1 leu2Δ0 met15Δ0 ura3-52,* for Rad6 protein level evaluation). The wild-type or mutant *BRE1* allele driven by the native *BRE1* promoter was inserted into pRS316. Strains were constructed with standard yeast genetic manipulation. Mutant strains were confirmed by PCR or sequencing.

## Replication stress/DNA damaging reagents sensitivity test

Sensitivity toward replication stress/DNA damaging reagents was tested using a spotting assay. A serial dilution of overnight yeast cultures were produced and 5 μl aliquots of the diluted culture were spotted onto YPD plates with the indicated replication stress/DNA damaging agents. Plates were incubated at 30 °C for 3–4 days before analysis.

## Analysis of DSB repair by ectopic recombination

To assess the cell survival due to successful DSB repair by ectopic recombination, cells were cultured in YEPD to the log phase and subsequently diluted and plated on YEPD or YEP-galactose plates, on which the HO expression is induced. Cells were allowed to grow at 30°C for 3–5 days. Survival rate is defined as the number of colonies grown on YEP-galactose plates divided by the number of colonies grown on YEPD plates times the fold of dilution.

## Cellular H2Bub1, Bre1, and Rad6 level analysis

Whole-cell yeast extracts were prepared using a trichloroacetic acid method as previously described (*Chen et al., 2012*). Samples were resolved with SDS–PAGE and transferred onto a Polyvinylidene

Difluoride (PVDF) membrane (Immobilon-P, Millipore) and proteins were detected with western blot. Flag-tagged H2B was detected with a mouse anti-FLAG antibody (F3165, Sigma-Aldrich, RRID: AB_259529, 1:3000 diluted) and the HRP-conjugated mouse IgGκ light chain binding protein (sc-516102, Santa Cruz Biotechnology, RRID: AB_2687626, 1:6000 diluted); 3xflag-tagged ScBre1 was detected with an anti-FLAG antibody (F1804, Sigma-Aldrich, RRID: AB_262044, 1:3000 diluted), TAP-tagged Rad6 was detected with an anti-TAP antibody (250067, Zen-BioScience, 1:6000 diluted); glyceraldehyde-3-phosphate dehydrogenase (GAPDH) was detected with an anti-GAPDH antibody (AC033, ABclonal, 1:50000 diluted).

## Acknowledgements

We thank scientists at the National Facility for Protein Science beamline BL19U1 at Shanghai Synchrotron Radiation Facility for setting up the beamline and assistance during diffraction data collection, the Large Equipment Sharing Platform at Tianjin Medical University for assistance with ITC and MS experiments, Dr. Jie Shen and the core facility at Tianjin Institute of Industrial Biotechnology, Chinese Academy of Sciences for assistance with SPR experiments, Drs Hang Zhang and Miaomiao Shen at Tianjin Medical University for their contribution in the early phases of the project. This work is supported by Natural Science Foundation of China (general grants 32271259, 32071205, and 31870769 to SX, 32070573 and 31872808 to XC).

## Additional information

### Funding

| Funder | Grant reference number | Author |
|---|---|---|
| National Natural Science Foundation of China | 32271259 | Song Xiang |
| National Natural Science Foundation of China | 32070573 | Xuefeng Chen |
| National Natural Science Foundation of China | 32071205 | Song Xiang |
| National Natural Science Foundation of China | 31870769 | Song Xiang |
| National Natural Science Foundation of China | 31872808 | Xuefeng Chen |

The funders had no role in study design, data collection, and interpretation, or the decision to submit the work for publication.

### Author contributions

Meng Shi, Jiaqi Zhao, Simin Zhang, Wei Huang, Mengfei Li, Xue Bai, Wenxue Zhang, Investigation; Kai Zhang, Investigation, Writing - review and editing; Xuefeng Chen, Resources, Supervision, Funding acquisition, Investigation, Visualization, Project administration, Writing - review and editing; Song Xiang, Conceptualization, Resources, Supervision, Funding acquisition, Investigation, Visualization, Writing - original draft, Project administration, Writing - review and editing

### Author ORCIDs

Song Xiang ⬤ http://orcid.org/0000-0001-9314-4684

### Decision letter and Author response

Decision letter https://doi.org/10.7554/eLife.84157.sa1
Author response https://doi.org/10.7554/eLife.84157.sa2

## Additional files

### Supplementary files
• MDAR checklist

### Data availability

Diffraction data and refined structures of crystal forms 1 and 2 of the KlBre1 RBD-Rad6 complex have been deposited into the protein data bank (https://www.rcsb.org), with accession codes 7W75 and 7W76, respectively. All data generated or analyzed during this study are included in the manuscript and supporting file; source data files are provided for Figures 1–5, Figure 1—figure supplement 3 and Figure 3—figure supplement 1.

The following datasets were generated:

| Author(s) | Year | Dataset title | Dataset URL | Database and Identifier |
|---|---|---|---|---|
| Xiang S | 2023 | Structural basis for the Rad6 activation by the Bre1 N-terminal domain | https://www.rcsb.org/structure/7W75 | RCSB Protein Data Bank, 7W75 |
| Xiang S | 2023 | Structural basis for the Rad6 activation by the Bre1 N-terminal domain | https://www.rcsb.org/structure/7W76 | RCSB Protein Data Bank, 7W76 |

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
