## [Editor Report]

This is a valuable structural study of partial Rad6 from K lactis in complex with Bre1 RBD domain. The structure provides detailed information on the interactions between these two proteins, which are validated by mutagenesis and functional studies. Overall, this is a well-executed study providing solid structural information useful to the field.

---

## [Decision Letter]

**Decision letter after peer review:**

Thank you for submitting your article "Structural basis for the Rad6 activation by the Bre1 N-terminal domain" for consideration by *eLife*. Your article has been reviewed by 3 peer reviewers, including Xiaobing Shi as the Reviewing Editor and Reviewer #1, and the evaluation has been overseen by Volker Dötsch as the Senior Editor.

Essential revisions:

In this manuscript, Shi et al. use X-ray crystallography to investigate the structural basis for Rad6 activation by Bre1. The structure of the K. lactis Bre1 RBD in complex with K. lactis Rad6 shows the binding of an asymmetric RBD dimer to Rad6. The structure is used to design mutants for testing how disruption of the Bre1 RBD-Rad6 interaction affects several Rad6 activities, including free ubiquitin chain synthesis, ubiquitin discharge, and active site accessibility. The authors also show that mutation of the RBD-Rad6 interface reduces H2Bub1 and DNA repair in yeast. Overall, this is a well-executed study providing nice structural information useful to the field. The following points need to be taken into consideration for revision.

1. Some statements are seemingly overstated based on only one experiment or one approach. Orthogonal approaches would be helpful to strengthen their conclusions. For instance, based on gel filtration profiles, the authors claim that substitutions do not alter the dimeric form of KlBre1-RBD. However, in Figure S5A, there exist some differences in the elution profiles of various RBD mutants and also in Figure 3D, the intensity of some RBD-ub mutants seems much weaker. Evidence from some other approaches, such as native SDS-PAGE or a protein cross-linking experiment, would be helpful to support their claims. Also, the salt concentration (1M) in ITC seems a bit too high. An alternative biophysical measurement would be helpful to validate the bindings.

2. The authors state "our data indicate Bre1 dimerization is required for the formation of a functional RBD". However, there is no experimental evidence supporting this claim. Could the authors design mutations to disrupt the Bre1 dimer and test the importance of dimerization for RBD function?

3. Figure 3B. The authors conclude that the reaction containing Bre1 RBD produces "significantly" more free ubiquitin chains than the reaction lacking the RBD. This experiment lacks quantitation, and the gel is not particularly convincing in showing that the RBD is having an effect. Is there an explanation for why the WT RBD is having a more noticeable effect in panel C compared to panel B? Does this have something to do with the use of mutant forms of ubiquitin and Rad6 in panel C?

4. Figure 3D-F. The authors conclude that the Bre1 RBD stimulates the rate of ubiquitin discharge from a Rad6-ub conjugate and that interface mutants slow this reaction. There are several issues with this experiment. First, the Rad6-ub conjugate is not being detected with equal efficiency in the control lacking the RBD and all other reactions. The authors speculate in the figure legend that the close migration of the RBD to the Rad6-ub conjugate is somehow interfering with the detection of the Rad6-ub conjugate by western blotting. This is an unusual explanation; however, it could be tested by running a longer gel or perhaps a gradient gel to better separate the RBD and Rad6-ub proteins. Second, often Rad6-ub discharge assays are quantified as the loss of the Rad6 conjugate band or the appearance of unconjugated Rad6. The levels of the Rad6-ub band, even in the presence of WT RBD, are not noticeably decreasing. This raises concerns about what is actually being measured when the authors use the di-ubiquitin band as a proxy for discharge. For some RBD mutants (e.g. R171E) the levels of the Rad6-ub conjugate appear to be increasing along with di-ubiquitin. Finally, rate constants are not calculated from the plots; it is difficult to discern whether there is any significant difference in the slopes of the lines. The authors' discussion of the limitations of this assay as done by Turco et al. could be written with greater clarity (lines 425-437).

5. The Ring domain of Bre1 in required for Bre1/Rad6-mediated H2Bub1 and subsequently biological function. To investigate the role of Bre1-RBD binding to Rad6, it is better to test the binding, Rad6-mediated ubiquitin chain production, and ubiquitin discharging of the E2~ubiquitin conjugate in the context of full-length KlBre1.

6. In the functional studies in yeast, are single point mutations (such as K31D, R36E, and R42E) sufficient to abolish Bre1 functions?

7. Some experiments lack proper controls or quantifications. See the minor points below.

---

## [Author Response]

Essential revisions:In this manuscript, Shi et al. use X-ray crystallography to investigate the structural basis for Rad6 activation by Bre1. The structure of the K. lactis Bre1 RBD in complex with K. lactis Rad6 shows the binding of an asymmetric RBD dimer to Rad6. The structure is used to design mutants for testing how disruption of the Bre1 RBD-Rad6 interaction affects several Rad6 activities, including free ubiquitin chain synthesis, ubiquitin discharge, and active site accessibility. The authors also show that mutation of the RBD-Rad6 interface reduces H2Bub1 and DNA repair in yeast. Overall, this is a well-executed study providing nice structural information useful to the field. The following points need to be taken into consideration for revision.1. Some statements are seemingly overstated based on only one experiment or one approach. Orthogonal approaches would be helpful to strengthen their conclusions. For instance, based on gel filtration profiles, the authors claim that substitutions do not alter the dimeric form of KlBre1-RBD. However, in Figure S5A, there exist some differences in the elution profiles of various RBD mutants and also in Figure 3D, the intensity of some RBD-ub mutants seems much weaker. Evidence from some other approaches, such as native SDS-PAGE or a protein cross-linking experiment, would be helpful to support their claims. Also, the salt concentration (1M) in ITC seems a bit too high. An alternative biophysical measurement would be helpful to validate the bindings.

We performed the suggested cross-linking experiment (Figure 1 figure supplement 3a), which shows that the wild type and substituted KlBre1 RBD can be efficiently cross-linked to dimers, supporting the notion that they are dimeric in solution. To validate the ITC data, we performed surface plasmon resonance (SPR) experiments. The SPR experiments show that KlBre1 RBD binds to KlRad6 with high affinity and the substitutions weakens the binding to various degrees, in line with the ITC and pull-down experiments. The intensities for the RBD-ub signal are much weaker for reactions with the K30A-, R35E- and R41E-substitued KlBre1 RBD in Figure 3D. These substitutions also severely inhibited the KlBre1 RBD-Rad6 interaction, suggesting that the Bre1 RBD-Rad6 interaction plays a critical role in ubiquitin discharging to Bre1 RBD. A similar observation is made with the full length KlBre1 (Author response image 1). We have changed the wording in several places in the manuscript to tune down our claims.

**Author response image 1. sa2fig1:** Biochemical characterization of the full length KlBre1. The full length KlBre1 gene was inserted into vector pET26B, with a strep-tag engineered to its N-terminus. The protein was expressed in *E. coli* BL21 Rosetta (DE3) cells and purified by strep-tactin (Smart Lifesciences), Heparin (Hitrap Heparin HP, GE healthcare) and gel filtration (Superose 6 10/300, GE heathcare) columns and stored in a buffer containing 20 mM Tris (pH 7.5), 200 mM sodium chloride and 2 mM DTT. (A) KlBre1-Rad6 pull down experiments. The reaction mixture contains 10 μM of KlBre1 and 3.2 μM of KlRad6 and the reactions were performed similar to the KlBre1 RBD-Rad6 pull-down experiments described in the manuscript. SDS PAGE analysis of KlBre1 precipitated with strep-tactin beads and co-precipitated KlRad6 is shown. Analysis of the input proteins are shown on the left. (B) KlRad6~ubiquitin discharging experiments. Reactions analyzed by non-reducing SDS PAGE followed by western blot for ubiquitin are presented. The reactions were performed similar to the discharging reactions with KlRad6 (S111L), ubiquitin (K0/I44A) and ubiquitin (I44A) presented in the manuscript. They were allowed to proceed for 0, 10, 20 and 40 minutes. When indicated, KlBre1 is added to a final concentration of 15 μM. (C) Ubiquitin chain production experiments. The reactions were performed similar to the ubiquitin chain production experiments with KlRad6 (S111L) and ubiquitin (I44A) presented in the manuscript. When indicated, KlBre1 was added to a final concentration of 18 μM. Reactions were allowed to proceed for 5, 10 and 20 minutes before analysis by western blot for ubiquitin.

2. The authors state "our data indicate Bre1 dimerization is required for the formation of a functional RBD". However, there is no experimental evidence supporting this claim. Could the authors design mutations to disrupt the Bre1 dimer and test the importance of dimerization for RBD function?

Our structure suggests that such mutations are difficult to design. The RBD dimer interface is very large, burying more than 8000 square Å of surface area (for comparison, the RBD-Rad6 interface buries 2000 square Å of surface area). Such a large interface suggests a very stable dimer, which may require a large number of point mutations at the interface to disrupt. These large number of point mutations may have additional effects, for instance disrupting the secondary structure elements. In addition, residues contributing to the Bre1 RBD-Rad6 interaction are located in regions that also contribute to the RBD dimerization, mostly at the N-terminal region of CC1 (Figure 1 figure supplement 1). Therefore, it is also difficult to design truncational mutations to disrupt the RBD dimer while keeping its Rad6 binding site. Due to these difficulties, we did not design mutations to disrupt the RBD dimer. We have tuned down our statement about the function of the RBD dimerization in paragraph 1 in the Discussion section.

3. Figure 3B. The authors conclude that the reaction containing Bre1 RBD produces "significantly" more free ubiquitin chains than the reaction lacking the RBD. This experiment lacks quantitation, and the gel is not particularly convincing in showing that the RBD is having an effect. Is there an explanation for why the WT RBD is having a more noticeable effect in panel C compared to panel B? Does this have something to do with the use of mutant forms of ubiquitin and Rad6 in panel C?

In the previous version of the manuscript, ubiquitin signal in Figure 3B was detected with SDS PAGE, whereas ubiquitin signal in Figure 3C was detected with western blot. The staining and scanning of the SDS PAGE gel may not be ideal and may contribute to the less apparent contrast. We have now replaced Figure 3B with a western blot analysis of the same reaction, which shows a clearer contrast. KlRad6 (S111L) and ubiquitin (I44A) were used in reactions presented in Figure 3C, these substitutions may alter the reaction rate. For instance, the I44A substitution in ubiquitin is located at the interface with the E1 enzyme during the ubiquitin charging reaction and may alter this reaction rate.

4. Figure 3D-F. The authors conclude that the Bre1 RBD stimulates the rate of ubiquitin discharge from a Rad6-ub conjugate and that interface mutants slow this reaction. There are several issues with this experiment. First, the Rad6-ub conjugate is not being detected with equal efficiency in the control lacking the RBD and all other reactions. The authors speculate in the figure legend that the close migration of the RBD to the Rad6-ub conjugate is somehow interfering with the detection of the Rad6-ub conjugate by western blotting. This is an unusual explanation; however, it could be tested by running a longer gel or perhaps a gradient gel to better separate the RBD and Rad6-ub proteins. Second, often Rad6-ub discharge assays are quantified as the loss of the Rad6 conjugate band or the appearance of unconjugated Rad6. The levels of the Rad6-ub band, even in the presence of WT RBD, are not noticeably decreasing. This raises concerns about what is actually being measured when the authors use the di-ubiquitin band as a proxy for discharge. For some RBD mutants (e.g. R171E) the levels of the Rad6-ub conjugate appear to be increasing along with di-ubiquitin. Finally, rate constants are not calculated from the plots; it is difficult to discern whether there is any significant difference in the slopes of the lines. The authors' discussion of the limitations of this assay as done by Turco et al. could be written with greater clarity (lines 425-437).

To search for a condition in which KlBre1 RBD does not interfere with the detection of the KlRad6~ubiquitin signal, we compared SDS PAGE and Western blot analysis of the same KlRad6~ubiquitin sample in the absence and presence of KlBre1 RBD (1-206) with 15%, 18% and 8~16% gradient gels (Figure 3 figure supplement 1F). We found that only in SDS PAGE with 18% gel KlBre1 RBD (1-206) and KlRad6~ubiquitin ran as distinct bands, but in all three cases KlBre1 RBD (1-206) weakens the signal of the KlRad6~ubiquitin conjugate in western blot. Therefore, western blot may not be a good technique to detect the KlRad6~ubiquitin signal in the presence of KlBre1 RBD (1-206) and such signal presented in Figure 3D is not reliable. We used SDS PAGE with 18% gel to detect the KlRad6~ubiquitin signal instead (Figure 3 figure supplement 1G-H). This experiment indicates that in the discharging reaction with KlRad6 (S111L), ubiquitin (I44A/K0) and ubiquitin (I44A) KlBre1 RBD (1-206) also stimulates the disappearance of the KlRad6~ubiqutin conjugate. The stimulation is reduced by substitutions in KlBre1 RBD inhibiting the KlBre1 RBD-Rad6 interaction, with the smallest reduction observed for the Q22A substitution and largest reductions observed for the K30A, R35E and R41E substitutions. In a similar experiment presented in Figure 3 figure supplement 1E, we found that in a discharging experiment with wild type KlRad6 and ubiquitin, KlBre1 RBD (1-184) also stimulated the KlRad6~ubiquitin discharge. We have calculated the slope of the reaction plots to represent the reaction rates (Figure 3G), which shows that the reaction with the wild type or Q22A-substituted KlBre1 RBD (1-206) are faster than reactions without KlBre1 RBD or with other KlBre1 RBD (1-206) variants. We have revised the discussion on the assays done by Turco et al. to make our points clearer.

5. The Ring domain of Bre1 in required for Bre1/Rad6-mediated H2Bub1 and subsequently biological function. To investigate the role of Bre1-RBD binding to Rad6, it is better to test the binding, Rad6-mediated ubiquitin chain production, and ubiquitin discharging of the E2~ubiquitin conjugate in the context of full-length KlBre1.

We have performed the suggested experiments (Author response image 1 attached to this document) but a conclusion cannot be drawn from the data whether the full length KlBre1 stimulates or inhibits Rad6’s activity. Our pull-down experiment indicated a clear KlBre1-Rad6 interaction, which is abolished by the R35E substitution in KlBre1 that also abolishes the KlBre1 RBD-Rad6 interaction (Author response image 1). Discharging reactions with KlRad6 (S111L), ubiquitin (I44A) and ubiquitin (I44A/K0) indicated that KlBre1 increases the KlRad6~ubiquitin conjugate discharge but decreases the rate of di-ubiquitin production and is ubiquitinated (Author response image 1). In free ubiquitin chain production experiments, we found that supplementing KlBre1 inhibited the production of free ubiquitin chains with molecular weights between 20~70 kDa but accelerated the production of ubiquitin chains with molecular weights above 70 kDa, which are presumably primarily attached to KlBre1 (Author response image 1). The effects of KlBre1 in ubiquitin discharging and ubiquitin chain formation were weakened by the R35E substitution in KlBre1 (Author response image 1). These data are similar to the data obtained from KlBre1 RBD. Together they suggest that in addition to ubiquitin discharging to ubiquitin (reaction 1), supplementing KlBre1 or KlBre1 RBD introduced a second reaction, ubiquitin discharging to KlBre1 or KlBre1 RBD (reaction 2). The amount of the reaction 2 product, ubiquitinated KlBre1 or KlBre1 RBD, are reduced by the R35E substitution in KlBre1 (Author response image 1), or the K30A, R35E and R41E substitutions in KlBre1 RBD (Figure 3D), suggesting that a strong interaction between KlBre1 or KlBre1 RBD and Rad6 is required for efficient reaction 2. Reaction 1 is relevant to Rad6’s activity and the H2Bub1 catalysis in vivo. It is not clear whether reaction 2 takes place in vivo. Since both reactions 1 and 2 compete for the Rad6~ubiquitin conjugate, reaction 2 is expected to inhibit reaction 1. To accurately assess the regulation of KlBre1 or KlBre1 RBD on Rad6’s activity represented by reaction 1, it is necessary to accurately determine the reaction kinetic parameters of reaction 2, which is difficult. Our data presented in the manuscript indicate that even with the inhibition of reaction 2, supplementing KlBre1 RBD increases the reaction 1 rate. This data suggests that KlBre1 RBD stimulates Rad6’s activity and the increase in reaction 1 rate is an underestimation of the stimulation. Here, we found that supplementing KlBre1 decreased the reaction 1 rate but increased the KlRad6~ubiquitin discharge. Without knowing the reaction kinetic parameters of reaction 2, it is not possible to determine whether the decrease in reaction 1 rate is primarily due to (1) an inhibition of KlRad6’s activity by KlBre1, or (2) an inhibition of reaction 1 by reaction 2, in which case KlBre1 could stimulate KlRad6’s activity to some extend and the overall effect is a decrease in the reaction 1 rate. Therefore, based on the data presented in Author response image 1, it is not possible to conclude whether KlBre1 stimulates or inhibits Rad6’s activity. Since a conclusion cannot be drawn from this data, we did not include it in the revised manuscript.

6. In the functional studies in yeast, are single point mutations (such as K31D, R36E, and R42E) sufficient to abolish Bre1 functions?

We tested the single substitutions and included the data in Figure 5. We found that the single point substitutions Q30A, K31D, R36E or R42E in Bre1 led to defects in H2B ubiquitination, HR repair and DNA damage response that are comparable to the defects caused by the *bre1* null mutant.

7. Some experiments lack proper controls or quantifications. See the minor points below.

We thank the reviewers for pointing them out and have revised the manuscript accordingly.